# Training Graph Neural Networks Subject to a Tight Lipschitz Constraint

Simona Juvină[1], Ana Neacșu[1,2], Jean-Christophe Pesquet[2], Corneliu Burileanu[1], Jérôme Rony[3] and Ismail Ben Ayed[3]

[1]National University of Science and Technology Politehnica Bucharest, SpeeD
[2]Université Paris-Saclay, Inria, CentraleSupélec, CVN
[3]ÉTS Montréal

**Reviewed on OpenReview:** `https://openreview.net/forum?id=KLojVqdj2y`

## Abstract

We propose a strategy for training a wide range of graph neural networks (GNNs) under tight Lipschitz bound constraints. Specifically, by leveraging graph spectral theory, we derive computationally tractable expressions of a tight Lipschitz constant. This allows us to propose a constrained-optimization approach to control the constant, ensuring robustness to adversarial perturbations. Unlike the existing methods for controlling the Lipschitz constant, our approach reduces the size of the handled matrices by a factor equal to the square of the number of nodes in the graph. We employ a stochastic projected subgradient algorithm, which operates in a block-coordinate manner, with the projection step performed via an accelerated iterative proximal algorithm. We focus on defending against attacks that perturb features while keeping the topology of the graph constant. This contrasts with most of the existing defenses, which tackle perturbations of the graph structure. We report experiments on various datasets in the context of node classification tasks, showing the effectiveness of our constrained GNN model.

## 1 Introduction

In recent years, graphs received increasing attention due to their effectiveness in accurately representing real-world data in a variety of fields, including social networks (Facebook, Twitter), citation networks (CiteSeer, PubMed), and biological networks (BioGRID). Node classification is one of the most frequent tasks when considering graph data. Considering a single graph, where labels are available only for a small subset of nodes, the aim is to predict the class labels of the remaining nodes. For example, in e-commerce, GNNs leverage the interaction between users and products, achieving highly accurate product advertising (Zhou et al., 2018). In citation networks, one may use GNNs to assign scientific articles to different topics (Kipf & Welling, 2016).

Being highly complex and non-linear structures, GNNs may be very sensitive to small perturbations of the input, which may result in a large deviation from the prediction. For secure applications, it is mandatory to control the effect of such perturbations. The Lipschitz constant is a well-known measure (Szegedy et al., 2014; Scaman & Virmaux, 2018) to characterize the robustness of neural networks to small perturbations. This constant provides an upper bound on the ratio between the output and input variations for a given distance. Thus, by controlling the Lipschitz constant, one can expect to increase the robustness of a network against adversarial perturbations.

In the context of GNNs, most of the existing adversarial attacks and defenses consider binary node features and focus on perturbing the adjacency matrix, not the node features. To the best of our knowledge, this work is the first to address the case of networks with continuous node features and fixed topology, *i.e.*, with a constant graph adjacency matrix. There is a breadth of applications (*e.g.*, transportation networks,

financial/commercial networks) where the scenario we tackle is relevant, although overlooked in the GNN literature. For example, most cutting-edge traffic control systems use GNNs to streamline traffic, by predicting the expected level of traffic in a given intersection or node based on information from its immediate neighbours and, to an extent, from the entire network, and directing traffic towards less congested paths. A malicious agent could disrupt this critical system by causing it to mislabel already congested intersections as being uncongested and directing even more traffic toward them. In addition, GNNs are used for electric power distribution, by predicting levels of power consumption and production, and also the electrical load on each node in the electrical grid. An attack that induces misclassification of a node as having a low power load could push the system to route more power through that node, resulting in an electrical overload. Also, in practice, inputs can be adversarial without third-party malicious intent. For example, in the case of an electrical network, there might be faulty sensors, or we might encounter erroneous readings. For economic or road networks, outliers might happen in some nodes, which may severely affect the performance of pre-trained models, with potentially harmful consequences.

In this work, we propose a learning scheme for training various GNN models under tight Lipschitz-bound constraints. We derive closed-form expressions of a tight Lipschitz constant which are easily computable. In addition, we propose a constrained optimization strategy to control this constant, ensuring robustness to adversarial perturbations[1]. Unlike the existing methods for controlling the Lipschitz constant, our approach reduces the size of the handled matrices by a factor of $K^2$, $K$ being the number of nodes in the graph. We propose a stochastic projected subgradient algorithm, which operates in a block-coordinate manner, with the projection step performed via an iterative proximal technique.

As we focus on defending against attacks that perturb node features, rather than graph structure, we introduce GNN attack formulations for this scenario. We focus on global attacks, *i.e.*, attacks that aim to cause misclassification of the entire test set. We employ white-box attacks to generate the strongest attack and use a simple budget constraint that enforces a maximum perturbation on the nodes in the test set. The attacks are performed after training, *i.e.*, evasion attacks. Our method is tested on several publicly available social networks and web graph datasets. We analyse how the networks trained under Lipschitz constraints behave when affected by various perturbation levels, as compared to the ones trained conventionally, demonstrating better performance in most situations.

The rest of the paper is organized as follows. Section 2 provides an overview of the related literature. Section 3 introduces our main contributions concerning the computation of a tight Lipschitz bound, exemplified for two classes of GNNs. Section 4 covers experimental methodology and details the results obtained when evaluating our networks against white-box adversarial attacks. Comparisons with different methods are made. In addition, we perform an analysis of memory and computational requirements. Section 5 is dedicated to concluding remarks, while Section 6 identifies some limitations of our work.

## 2 Related Work

With the advancement of high-performance GNNs, robustness issues arise. In the context of graph-structured inputs, attacks may perturb the features, the adjacency matrix, or both simultaneously. However, given the intricacy of structural information, the majority of current attacks and defenses concentrate on perturbations of the adjacency matrix rather than the features.

### 2.1 Adversarial attacks of GNNs models

The authors of Dai et al. (2018) propose three methods for different attack settings, addressing only graph-structure perturbations, and covering inductive and transductive classification. The first method, *RL-S2V*, learns a generalizable attack policy over the graph structure while only requiring the prediction labels from the target classifier, being suitable for black-box attacks and transfer. *GradArgmax* is a white-box attack, which performs a greedy selection based on gradient information, and *GeneticAlg* is a black-box attack employing a genetic algorithm. In addition, the authors point out that the effect of the attack algorithm can be reduced by dropping edges randomly during training. In Bojchevski & Günnemann (2019), the authors introduce the

---

[1]A full PyTorch implementation is available at https://github.com/simona-juvina/lipschitz-gnn.

first study on adversarial perturbations for unsupervised embeddings. They propose a white-box attack for the family of methods based on random walks, where an attacker can add or remove edges in the original graph within a given budget. The attack is exemplified for two different scenarios: node classification and link prediction, in a local setting (affecting a single node or a single edge, respectively). Additionally, they show that the attacks generated for one model can generalize for different models and are successful even in scenarios where the attacker is restricted to specific nodes. In Zügner et al. (2018), the authors published the first study on adversarial attacks for attributed graphs, focusing on models based on graph convolution. *Nettack* is a white-box, non-adaptive attack that targets both the node features and the graph structure, in a local setting. Given a budget constraint, the attack greedily flips the best edges of the graph considering a linearized GCN. Additionally, the study distinguishes between attacker nodes and target nodes, which gives rise to two approaches, *i.e.*, direct attacks and influencer attacks. The experiments show that the attack is effective in drastically affecting the classification accuracy even with small perturbations. Furthermore, *Nettack* is transferable to other node classification models and unsupervised approaches. The study in Zügner & Günnemann (2019) focuses on a poisoning attack that targets global node classification (*Metattack*), where the attacker has full knowledge about the data, but no information about the model or its trained weights. The attack aims to modify the graph structure using a meta-gradient backpropagation procedure and adjusts the adjacency matrix of the graph as a hyperparametric problem. Similarly to *Nettack*, *Metattack* is a greedy multistep attack that flips a single edge in each iteration, up to a predefined budget, and uses a linearized Graph Convolutional Network (GCN). Only a few iterations are required for this attack to be effective.

## 2.2 Defense mechanisms for GNN models

The study in Wu et al. (2019) investigated both attack and defense techniques. The proposed attack uses integrated gradients to measure the importance of perturbing particular nodes and edges, and then flips the most influential ones. The defense method is based on improving the graph structure (prior to any message passing), by filtering out all edges that connect nodes that exhibit a low Jaccard similarity. This step does not significantly reduce the performance of the network while making the graph network more robust against targeted attacks. *ProGNN* (Jin et al., 2020) is another defense designed to improve the graph structure by jointly learning the parameters of the GNN and the adjacency matrix. The defense aims to reconstruct the original clean graph by preserving the properties of the original adjacency matrix (*e.g.*, low-rankness, sparsity, and feature smoothness). The results reported on real-world data demonstrate that *ProGNN* achieves better performance compared to state-of-the-art defense methods under *Metattack*, *Nettack*, and random attacks. The authors of Zhang & Zitnik (2020) proposed *GNNGuard*, a method to defend against a variety of evasion attacks that perturb the graph structure. The algorithm assigns higher weights to edges connecting nodes that exhibit a higher cosine-similarity, while adaptively weighting down edges that might be adversarial. *GNNGuard* achieves better robustness against both targeted and untargeted attacks compared to previous defenses. *RGCN* (Zhu et al., 2019) is a probabilistic defense technique aiming to improve the architecture of the network by representing the hidden states of all convolutional layers as Gaussian distributed. In addition, for each layer, the weights of node neighbourhoods are adjusted based on their variances.

The authors of Entezari et al. (2020) propose one of the few defense methods that applies to both the adjacency matrix and the binary feature matrix. The defense involves a low-rank approximation of graph data in order to filter out high-frequency noise introduced by adversarial attacks. The conducted experiments demonstrate the robustness of the proposed defense against *Nettack* perturbations. However, the recent study in Mujkanovic et al. (2022) suggests a limitation in the evaluation, as the paper does not include an assessment for feature-only attacks, and, given the strong bias of *Nettack* toward structural perturbations, their results may not confirm feature robustness. *The authors of Mujkanovic et al. (2022) pointed out the lack of studies that defend against feature perturbations, which is the objective of this work.*

Randomized smoothing has also emerged as an effective approach for enhancing the robustness of GNNs against adversarial perturbations. Bojchevski et al. (2020) tackled the challenge of achieving certified robustness against structural perturbations in graphs by leveraging sparsity. By exploiting graph connectivity and the localized nature of graph features, the method reduces the computational complexity of generating robustness certificates while maintaining high accuracy. Scholten et al. (2022) proposed a gray-box certification method for GNNs to defend against adversaries with full control over multiple node features. By employing

random edge deletion and node masking, they analysed the probability of adversarially controlled nodes transmitting messages to target nodes. Experimental results on binary-featured datasets show that their approach outperforms previous smoothing-based certificates, providing improved robustness guarantees for GNNs.

### 2.3 Controlling the Lipschitz constant

To the best of our knowledge, the existing literature on the Lipschitz behavior of GNNs is relatively limited. The authors of Dasoulas et al. (2021) introduced a normalization layer for basic attention-based GNNs, enforcing Lipschitz regularity. While this constraint facilitates building deep GNN architectures, the study did not investigate the robustness of these models against adversarial attacks. The work in Arghal et al. (2022) focused on GNNs where the convolution is expressed in the frequency domain. They suggested controlling the Lipschitz constant of GNN filters with respect to the node attributes and extending the formulation to dynamic graphs. Finally, Zhao et al. (2021) proposed an attack-agnostic, graph-adaptive defense against adversarial samples for GNNs. The method relies on a separable Lipschitz bound, constraining both the weight and gradient norms of each layer to 1. This corresponds to a Lipschitz bound looser than the one we propose in this work. Furthermore, the attacks considered in Zhao et al. (2021) perturb only the adjacency matrix, not the features.

## 3 Proposed Approach

### 3.1 Background and preliminaries

For the task of (semi-supervised) node classification on a single large graph with continuous node features, $G = (V, E)$ represents the (directed) graph, where $V = \{v_1, \ldots, v_K\}$ denotes the set of nodes and $E \subset V^2$ represents the edges. Each node $v \in V$ is associated with a vector $x_v \in \mathbb{R}^{N_0}$ representing continuous features and a label $c_v \in \{1, \ldots, C\}$, with $C$ the number of classes. For every node $v$, we define its neighborhood $\mathcal{N}(v)$ as the set of nodes $u$ such that there exists an edge going from $u$ to $v$. We consider a (generic) graph convolutional operator (Hamilton et al., 2017b; Morris et al., 2019) at node $v \in V$ and at layer $i \in \{1, \ldots, m\}$:

$$y_v^{(i)} = R_i\Big(w_0^{(i)} y_v^{(i-1)} + w_1^{(i)} \sum_{u \in \mathcal{N}(v)} \rho_{v,u} y_u^{(i-1)} + b_v^{(i)}\Big), \tag{1}$$

where $y_v^{(i)}$ is a vector of embeddings in $\mathbb{R}^{N_i}$, $b_v^{(i)}$ is a bias vector in $\mathbb{R}^{N_i}$, $w_0^{(i)}$ and $w_1^{(i)}$ are weight matrices in $\mathbb{R}^{N_i \times N_{i-1}}$, and $R_i$ is an activation operator from $\mathbb{R}^{N_i}$ to $\mathbb{R}^{N_i}$. Throughout the paper, we will make the mild assumption, satisfied by ReLU and sigmoid activations, that $R_i$ is 1-Lipschitz and that it applies componentwise, except possibly for the last layer (where a multivariate operator such as softmax might be used). For every $u \in \mathcal{N}(v)$, $\rho_{u,v} \in \mathbb{R}$ represents the weight between nodes $v$ and $u$. For the input, we set $N_0$ to the dimension $N$ of the feature space and $(\forall v \in V)\ y_v^{(0)} = x_v$, and for the output we set $N_m = C$.

We define the (weighted) adjacency matrix of the graph $G$ as $A^{\boldsymbol{\epsilon}} = (A^{\boldsymbol{\epsilon}}_{u_k, v_\ell})_{1 \leq k, \ell \leq K} \in \mathbb{R}^{K \times K}$ where, for every $(u, v) \in V^2$,

$$A^{\boldsymbol{\epsilon}}_{u,v} = \varepsilon_{u,v} \text{ if } u \in \mathcal{N}(v), \qquad 0 \text{ otherwise}, \tag{2}$$

and $\varepsilon_{u,v}$ is the positive weight of the edge from node $v$ to $u$.

Equation (1) can be expressed under the vector form:

$$\boldsymbol{y}^{(i)} = \boldsymbol{R}_i(\boldsymbol{W}_0^{(i)} \boldsymbol{y}^{(i-1)} + \boldsymbol{W}_1^{(i)} \boldsymbol{y}^{(i-1)} + \boldsymbol{b}^{(i)}), \tag{3}$$

with $\boldsymbol{R}_i \colon \mathbb{R}^{KN_i} \to \mathbb{R}^{KN_i}$, $\boldsymbol{y}^{(i)} \in \mathbb{R}^{KN_i}$, and $\boldsymbol{b}^{(i)} \in \mathbb{R}^{KN_i}$. Hereabove, $\boldsymbol{W}_0^{(i)}$ and $\boldsymbol{W}_1^{(i)}$ are $KN_i \times KN_{i-1}$ real-valued matrices, expressed as

$$\boldsymbol{W}_0^{(i)} = \mathrm{Id}_K \otimes w_0^{(i)}, \qquad \boldsymbol{W}_1^{(i)} = M \otimes w_1^{(i)}, \tag{4}$$

where $\otimes$ denotes the standard matrix Kronecker product. In the second part of Equation (4), the matrix $M \in \mathbb{R}^{K \times K}$ may take various forms, as detailed below.

Consider the following $K \times K$ degree matrix: $D = \text{Diag}[\text{d}(v_1), \ldots, \text{d}(v_K)]$. One may consider a number of variations of this model. In Equation (1), we may have:

- $\rho_{v,u} = \frac{1}{\text{d}(u)}$, which amounts to setting $M = A^\top D^{-1}$.

- $\rho_{v,u} = \frac{1}{\sqrt{\text{d}(v)\text{d}(u)}}$, which amounts to setting $M = D^{-1/2}A^\top D^{-1/2}$.

- $\rho_{v,u} = \varepsilon_{v,u}$, which amounts to setting $M = (A^{\boldsymbol{\varepsilon}})^\top$.

- $\rho_{v,u} = \frac{\varepsilon_{v,u}}{\sqrt{\text{d}^{\boldsymbol{\varepsilon}}(v)\text{d}^{\boldsymbol{\varepsilon}}(u)}}$, where $\text{d}^{\boldsymbol{\varepsilon}}(v) = \sum_{u \in \mathcal{N}(v)} \varepsilon_{v,u}$ and $A^{\boldsymbol{\varepsilon}} \geq 0$. This amounts to setting $M = (D^{\boldsymbol{\varepsilon}})^{-1/2}(A^{\boldsymbol{\varepsilon}})^\top (D^{\boldsymbol{\varepsilon}})^{-1/2}$, where $D^{\boldsymbol{\varepsilon}}$ is the diagonal matrix with diagonal terms $(\text{d}_\varepsilon(v_k))_{1 \leq k \leq K}$.

It is easy to notice that Model (3) can be written more concisely as follows:

$$(\forall i \in \{1, \ldots, m\}) \quad \boldsymbol{y}^{(i)} = \boldsymbol{R}_i(\boldsymbol{W}^{(i)}\boldsymbol{y}^{(i-1)} + \boldsymbol{b}^{(i)}), \tag{5}$$

where

$$\boldsymbol{W}^{(i)} = \boldsymbol{W}_0^{(i)} + \boldsymbol{W}_1^{(i)} = \text{Id}_K \otimes w_0^{(i)} + M \otimes w_1^{(i)}. \tag{6}$$

In Equation (5), we recognize the generic form of a feed-forward network. One of the main points is that, although weight matrix $\boldsymbol{W}^{(i)}$ may have a large size, it is usually very sparse. The zero elements of this matrix may be identified by computing

$$\boldsymbol{\Gamma}^{(i)} = \underbrace{(A^\top + \text{Id}_K)}_{K \times K} \otimes \underbrace{(1_{N_i}1_{N_{i-1}}^\top)}_{N_i \times N_{i-1}} \in \mathbb{R}^{(KN_i) \times (KN_{i-1})}, \tag{7}$$

where $1_{N_i} = [1, \ldots, 1]^\top \in \mathbb{R}^{N_i}$. Indeed, the zero elements of $\boldsymbol{\Gamma}^{(i)}$ are located at the same positions as the elements of $\boldsymbol{W}^{(i)}$, which need to be set to zero. The positions of these elements being easily identified, one could use existing methods for controlling the Lipschitz constant of standard networks (Cisse et al., 2017; Fazlyab et al., 2019; Neacşu et al., 2021) to train a robust GNN based on Equation (5). However, such an approach turns out to be computationally expensive for large graphs, *i.e.*, when the number $K$ of nodes is large. Furthermore, it is prohibitive in terms of memory requirements, even for small datasets. It requires dealing with large-size matrices of size $KN_i \times KN_{i-1}$, to perform spectral normalization of these matrices[2] and, at the same time, to impose sparse algebraic constraints. We propose a scalable approach, which reduces the size of the handled matrices by a factor $K^2$, being agnostic to the size of the graph.

## 3.2 Robustness quantification

The forward pass through an $m$-layer fully connected neural network $T$ can be viewed as the composition of functions given by Equation (5). If the graph input feature vector $\boldsymbol{x} \in \mathbb{R}^{KN_0}$ is perturbed by $\boldsymbol{\delta} \in \mathbb{R}^{KN_0}$, the effect can be quantified by the following inequality:

$$\|T(\boldsymbol{x} + \boldsymbol{\delta}) - T(\boldsymbol{x})\| \leq \theta\|\boldsymbol{\delta}\|, \tag{8}$$

where $\|\cdot\|$ is the Euclidean norm and $\theta$ denotes a Lipschitz constant of the model. The pioneering work of Szegedy et al. (2014) points out that, if $\theta_i$ denotes the Lipschitz constant of layer $i$, a Lipschitz constant of the full network can be derived as

$$\theta = \prod_{i=1}^{m} \theta_i. \tag{9}$$

However, such a separable bound might be loose (Scaman & Virmaux, 2018).

We now state our main result, which provides an easy-to-compute non-separable tight upper bound. The proof of this result is grounded on graph spectral theory and some of the results in Combettes & Pesquet (2020). The details of the proof are provided in Appendix A.2.

---

[2]Such operations hardly benefit from the sparse matrix structure or its expression as a sum of Kronecker products.

**Theorem 3.1.** *Consider our model of a generic graph convolutional neural network. Assume that matrix $M$ in Equation (4) is symmetric (corresponding to an undirected graph) with non-negative elements. Let $\lambda_K \geq 0$ be its maximum eigenvalue. Assume that, for every $i \in \{1, \ldots, m\}$, matrices $w_0^{(i)}$ and $w_1^{(i)}$ have non-negative elements, $w_0^{(i)} \geq 0$ and $w_1^{(i)} \geq 0$. Let*

$$(\forall \mu \in \mathbb{R}) \quad \varphi(\mu) = \|(w_0^{(m)} + \mu w_1^{(m)}) \cdots (w_0^{(1)} + \mu w_1^{(1)})\|_\mathrm{S}, \tag{10}$$

*where $\|.\|_\mathrm{S}$ denotes the spectral norm. Then, a Lipschitz constant of the network is given by*

$$\vartheta = \varphi(\lambda_K). \tag{11}$$

*In addition, $\vartheta$ is the optimal Lipschitz constant when there is no bias and ReLU activations are used at all layers.*

Evaluations of the differences between the loose and tight estimations can be found in Appendix B. In the following, we will study two special cases of this formulation corresponding to two of the most widely used GNNs, namely Graph Convolutional Network (GCN) (Kipf & Welling, 2016) and GraphSAGE (Hamilton et al., 2017a). Then, we will propose a strategy to control the Lipschitz constant $\theta$ during the training of our model. It is worth noting that, not only do our results apply to other choices for the adjacency matrix, but they do not assume that the network is linear. It can include arbitrary nonexpansive activation functions and biases.

## 3.3  Robust GraphSAGE

We consider a symmetric version of the GraphSAGE layer (Hamilton et al., 2017a), using the mean operator as the aggregator function:

$$y_v^{(i)} = R_i \Big( w_0^{(i)} y_v^{(i-1)} + w_1^{(i)} \frac{\sqrt{\mathrm{d}(v)}}{|\mathcal{N}(v)|} \sum_{u \in \mathcal{N}(v)} \widetilde{y}_u^{(i-1)} + b_v^{(i)} \Big), \tag{12}$$

where, for every $u \in V$, $\widetilde{y}_u^{(i-1)} = y_u^{(i-1)} / \sqrt{\mathrm{d}(u)}$. This model is a particular case of Equation (1). In the case of an undirected graph, matrix $M \in \mathbb{R}^{K \times K}$ is expressed as

$$M = D^{-1/2} A^\top D^{-1/2} = D^{-1/2} A D^{-1/2}. \tag{13}$$

This matrix being symmetric, following Theorem 3.1 and Appendix A.3(ii) we have

$$\vartheta = \varphi(1) = \|(w_0^{(m)} + w_1^{(m)}) \cdots (w_0^{(1)} + w_1^{(1)})\|_\mathrm{S}. \tag{14}$$

## 3.4  Robust Graph Convolutional Network (GCN)

The GCN model (Kipf & Welling, 2016) satisfies Equation (5), with

$$\boldsymbol{W}^{(i)} = \widetilde{M} \otimes w_1^{(i)}, \tag{15}$$

and

$$\widetilde{M} = \widetilde{D}^{-1/2} \widetilde{A}^\top \widetilde{D}^{-1/2}. \tag{16}$$

$\widetilde{A} = \mathrm{Id}_K + A = (\widetilde{A}_{k,\ell})_{1 \leq k, \ell \leq K}$ is the adjacency matrix of the undirected graph $G$ with added self-connections, $\widetilde{D}$ is the diagonal matrix with diagonal elements $\widetilde{d}_k = \sum_{\ell=1}^K \widetilde{A}_{k,\ell}$, $k \in \{1, \ldots, K\}$, and $w_1^{(i)}$ is the weight matrix corresponding to the $i$-th layer. Since $\widetilde{A}$ is a symmetric matrix with non-negative elements, the results in the previous sections can be extended. As shown in Appendix A.4, when for every $i \in \{1, \ldots, m\}$ $w_1^{(i)} \geq 0$, we obtain the following Lipschitz constant:

$$\vartheta = \|w_1^{(m)} \cdots w_1^{(1)}\|_\mathrm{S}. \tag{17}$$

### 3.5 Constraint set

Let us summarize the constraints we need to deal with to train a robust GNN. For simplicity, we focus on GraphSAGE. The extensions to other GNNs based on Model (1) are straightforward.

(i) *Non-negativity.* To ensure the validity of the proposed tight bound for the Lipschitz constant we impose that, for each layer $i \in \{1, \ldots, m\}$, the weight matrices are non-negative valued, so leading to the closed convex constraint set

$$\mathcal{D}^{(i)} = \{(w_0^{(i)}, w_1^{(i)}) \in (\mathbb{R}^{N_i \times N_{i-1}})^2 \mid w_0^{(i)} \geq 0 \text{ and } w_1^{(i)} \geq 0\}. \tag{18}$$

(ii) *Lipschitz bound constraint.* Considering $\overline{\vartheta}$ as the target maximum Lipschitz constant of the network, we need to impose the following constraint on the spectral norm of the weight matrices:

$$\|(w_0^{(m)} + w_1^{(m)}) \cdots (w_0^{(1)} + w_1^{(1)})\|_{\text{S}} \leq \overline{\vartheta}. \tag{19}$$

This bound aims at improving the robustness of the network against adversarial attacks. Note that this constraint is not convex. However, if $(w_0^{(i')}, w_1^{(i')})_{i' \neq i}$ are given and we just optimize $(w_0^{(i)}, w_1^{(i)})$, we deal with a convex constraint.

These two constraints must be handled simultaneously during training.

### 3.6 Dealing with the Constraint Set

Calculating the Lipschitz constant of a neural network is generally NP-hard (Scaman & Virmaux, 2018), although computing a tight estimate of it is of paramount importance for efficient control of the robustness during the training phase. Theorem 3.1 allows us to circumvent this computational bottleneck. To impose a bound on the Lipschitz constant, we resort to a stochastic projected subgradient strategy that operates in a block-coordinate manner. Although the bound on the product in Constraint 19 defines a non-convex function, we are dealing with a multi-convex problem, which justifies the use of a block-coordinate approach.

Let us describe more precisely how we proceed by focusing on the GraphSAGE case.

To update the weights, we employ a projected stochastic gradient optimizer or an Adam variant of it (Kingma & Ba, 2015). At epoch $t > 0$, we denote by $w_{0,t}^{(i)}$ and $w_{1,t}^{(i)}$ the estimated weight matrices for layer $i$ and define $W_t^{(i)} = \begin{bmatrix} w_{0,t}^{(i)} \\ w_{1,t}^{(i)} \end{bmatrix}$. When optimizing this matrix at epoch $t + 1$, Constraint (19) is re-expressed as

$$\mathcal{C}_t^{(i)} = \left\{ W_t^{(i)} \in \mathbb{R}^{(2N_i) \times N_{i-1}} \left| \left\| A_t^{(i)} W_t^{(i)} B_t^{(i)} \right\|_{\text{S}} \leq \overline{\vartheta} \right. \right\}, \tag{20}$$

where matrices $A_t^{(i)}$ and $B_t^{(i)}$ represent the product of the estimated matrices $(w_{0,t}^{(j)} + w_{1,t}^{(j)})_{j \neq i}$ for the next and previous layers, respectively:

$$\begin{aligned} A_t^{(i)} &= [I_{N_m} \ I_{N_m}] W_t^{(m)} \cdots [I_{N_{i+1}} \ I_{N_{i+1}}] W_t^{(i+1)} [I_{N_i} \ I_{N_i}], \\ B_t^{(i)} &= [I_{N_{i-1}} \ I_{N_{i-1}}] W_{t+1}^{(i-1)} \cdots [I_{N_1} \ I_{N_1}] W_{t+1}^{(1)}. \end{aligned} \tag{21}$$

Accordingly, for the first layer with $i = 1$, we have $B_t^{(i)} = I_{N_0 \times N_0}$, and for the last one with $i = m$, $A_t^{(i)} = [I_{N_m} \ I_{N_m}]$. Note that the previous layers have already been updated (justifying the use of index $t + 1$), when we deal with layer $i$.

Recalling that $\mathcal{D}_i$ denotes the non-negativity constraint (Constraint 18), our training procedure performs iteratively a gradient step on $W_t^{(i)}$, and then projects the result $\widehat{W}_t^{(i)}$ onto the set $\mathcal{S}_t^{(i)} = \mathcal{D}^{(i)} \cap \mathcal{C}_t^{(i)}$, for

each layer $i \in \{1, \ldots, m\}$ and at each epoch $t + 1$. Note that this intersection defines a closed convex set, which is one of the main advantages of the proposed block coordinate approach operating layerwise. The projection onto the intersection of $\mathcal{D}^{(i)}$ and $\mathcal{C}_t^{(i)}$ has no closed-form expression and, for computing it, we employ an iterative algorithm. More specifically, we make use of an instance of the accelerated iterative dual forward-backward (DFB) algorithm as proposed in Neacşu et al. (2024). The projection algorithm is detailed in Algorithm 1.

---

**Algorithm 1** Accelerated version of the Dual Forward-Backward (DFB) algorithm

**Let:** $Y_0 \in \mathbb{R}^{N_m \times N_0}$
**Set:** $\gamma = 1/(\|A_t^{(i)}\|_{\mathrm{S}} \|B_t^{(i)}\|_{\mathrm{S}})^2$
**Set:** $\alpha \in ]2, +\infty[$
**for** $l = 0, 1, \ldots$ **do**
  $\eta_l = \frac{l}{l+1+\alpha}$
  $Z_l = Y_l + \eta_l(Y_l - Y_{l-1})$
  $V_l = \mathrm{proj}_{\mathcal{D}_i}(\widehat{W}_t^{(i)} - A_t^{(i)\top} Z_l B_t^{(i)\top})$          // Truncate negative values
  $\widetilde{Y}_l = Z_l + \gamma A_t^{(i)} V_l B_t^{(i)}$
  $Y_{l+1} = \widetilde{Y}_l - \gamma \, \mathrm{proj}_{\mathcal{B}(0,\overline{\vartheta})}(\gamma^{-1}\widetilde{Y}_l)$
**end for**
**Return:** $V_l = \mathrm{proj}_{\mathcal{S}_t^{(i)}} \widehat{W}_t^{(i)}$

---

Hereabove, $\mathrm{proj}_{\mathcal{D}_i}$ represents the projection onto the cone of non-negative valued matrices, whereas $\mathrm{proj}_{\mathcal{B}(0,\overline{\vartheta})}$ stands for the projection onto the spectral ball of radius $\overline{\vartheta} > 0$. The latter projection is easily computed by a singular value decomposition.

## 4 Experimental Approach and Results

### 4.1 Experimental Methodology

To evaluate the effectiveness of the proposed method for training robust graph networks, we conducted experiments on various publicly available graph datasets comprising real-world social networks and web graphs. The considered datasets are Facebook (Rozemberczki et al., 2021), GitHub (Rozemberczki et al., 2021), LastFM Asia (Rozemberczki & Sarkar, 2020), and Deezer Europe (Rozemberczki & Sarkar, 2020). These graphs are attributed, allow for binary and multi-class node classification, and are variable in size and density. More details about the datasets are provided in Appendix C.1.

For all datasets, we conducted experiments using our two proposed networks: robust GCN and robust GraphSAGE. We report mean accuracy with standard errors calculated from 10 seeded splits, with a training-validation-testing split ratio of 60%, 20%, and 20%, respectively. For both architectures, we trained models subject to various Lipschitz bound constraints ($\overline{\vartheta} \in [1, 30]$), to find the best trade-off between robustness and performance. Unless otherwise stated, we consider networks with $m = 3$ layers and hidden feature dimension $N_i = 16$ for $i \in \{1, 2\}$. Furthermore, in a part of our studies, the influence of the depth of the network on the overall behavior is evaluated by varying the number of layers $m \in [2, 7]$ (see discussion in Appendix D.5). The rectified linear unit function (ReLU) activation function is used for intermediate layers, and Adam optimizer (Kingma & Ba, 2015) with $\ell_2$ penalty is employed to optimize the model parameters. The general formulation in Equation (1) allows for a distinctive bias parameter for each node. However, in our experiments, a single bias vector is used for the entire layer and applied element-wise to the features of all nodes, which is the standard practice. Our method is based on the assumption of non-negativity. Although this constraint is intrinsic to our approach, from our experiments, we found that the non-negativity requirement affects the accuracy only by a small margin, for these datasets. A comparison in accuracy between training a network with non-negative weights and conventional training is presented in Table 1. It is worth highlighting that while the weights within the network are nonnegative, the inclusion of signed biases introduces arbitrary signs in the outputs after each layer. Consequently, this allows for the activation function to be triggered.

Table 1: Mean accuracy ± standard deviation of the baseline and non-negative constrained models on all datasets from 10 seeded splits.

|  |  | Facebook | LastFM | GitHub | Deezer Europe |
|---|---|---|---|---|---|
| GCN | Baseline | 94.0+/-0.0 | 87.3+/-0.7 | 86.9+/-0.6 | 60.5+/-0.5 |
|  | Pos. constr. | 93.2+/-0.4 | 85.5+/-0.7 | 86.0+/-0.8 | 59.1+/-1.2 |
| GraphSAGE | Baseline | 93.3+/-0.5 | 85.5+/-0.8 | 86.5+/-0.5 | 65.2+/-0.4 |
|  | Pos. constr. | 92.2+/-0.6 | 85.3+/-0.7 | 85.9+/-0.7 | 64.4+/-0.5 |

Most of the existing adversarial attacks and defenses for GNNs primarily focus on perturbing the adjacency matrix rather than the node features (Mujkanovic et al., 2022). From our knowledge, not all the existing works that address Lipschitz constraints on GNNs consider graph convolutional operators. Arghal et al. (2022) tackle Lipschitz normalization for self-attention layers, which is not directly comparable to our approach. Additionally, these papers only provide numerical results for datasets with binary features, and their code is not always publicly available, making it challenging to make meaningful comparisons. Thus, we compare the performance of our robust training approach with two standard methods, namely Adversarial Training (AT) (Madry et al., 2018) and Randomized Smoothing (RS) (Cohen et al., 2019). Adversarial training involves generating adversarial samples using the Auto-PGD (APGD) attack (Croce & Hein, 2020) with a Difference of Logits (DL) loss and training models for different perturbation levels $\varepsilon \in [0, 1000]$, which represent the $\ell_2$ norm of the perturbation on the entire test set. For RS we consider the method described in Cohen et al. (2019) and conduct experiments using various Gaussian noise levels $\sigma \in [0, 2]$, applied element-wise to each feature vector of the input data. Finally, in the case of GCN networks, we also compare our approach with Spectral Normalization (SN) (Miyato et al., 2018) and methods aimed to defend against graph structure perturbation: GCN-Jaccard (Wu et al., 2019) and SVD-GCN (Entezari et al., 2020), which focus on improving the structure of the graph, and RGCN (Zhu et al., 2019), which utilizes adaptive aggregation. Further details on the training strategies can be found in Appendix C.2.

When training constrained models and imposing a lower Lipschitz constraint than that of the baseline model $\bar{\vartheta} < \theta_{\text{model}}$, the standard accuracy decreases, but the model exhibits greater robustness. For instance, consider the accuracy of GCN models trained by varying constraint values $\bar{\vartheta}$ on the Facebook dataset (refer to Table 2). As we tighten the constraint, the drop in accuracy becomes more significant, but the model's ability to withstand adversarial perturbations improves compared to the conventionally trained model. Figure 1 exemplifies this behavior on different datasets, for both GCN and GraphSAGE architectures, presenting the accuracy of the baseline and constrained models with different $\bar{\vartheta} < \theta_{\text{model}}$, while varying the $\ell_2$ norm of the test set perturbation. This trade-off pattern holds for the other methods presented in this paper with respect to their specific parameters, as the perturbation level increases. Acknowledging this inherent trade-off between accuracy and robustness in all methods, we opted for choosing one model from each technique for the purpose of comparison. Thus, we selected the model with the largest area under the accuracy-robustness curve, while ensuring that it exhibits a decrease in relative accuracy on the validation dataset of less than 3% when compared to the conventionally trained model (when possible).

Table 2: Mean accuracy ± standard deviation of GCN models trained under different Lipschitz constraints on the Facebook dataset from 10 seeded splits.

|  | $\theta_{\text{model}} = 51.99$ | $\bar{\vartheta} = 30$ | $\bar{\vartheta} = 25$ | $\bar{\vartheta} = 20$ | $\bar{\vartheta} = 15$ | $\bar{\vartheta} = 10$ | $\bar{\vartheta} = 5$ | $\bar{\vartheta} = 3$ |
|---|---|---|---|---|---|---|---|---|
| Acc. | $94.0 \pm 0.0$ | $93.1 \pm 0.4$ | $92.9 \pm 0.5$ | $92.2 \pm 0.4$ | $91.7 \pm 0.5$ | $90.8 \pm 0.6$ | $87.9 \pm 0.7$ | $84.3 \pm 1.1$ |

## 4.2 Adversarial robustness evaluation

In this section, we evaluate the efficiency of our robust training approach against white-box adversarial attacks and compare it with conventional training, AT, and RS. Although there are not many works studying adversarial attacks on graph features (Mujkanovic et al., 2022), we can use the classical Projected Gradient

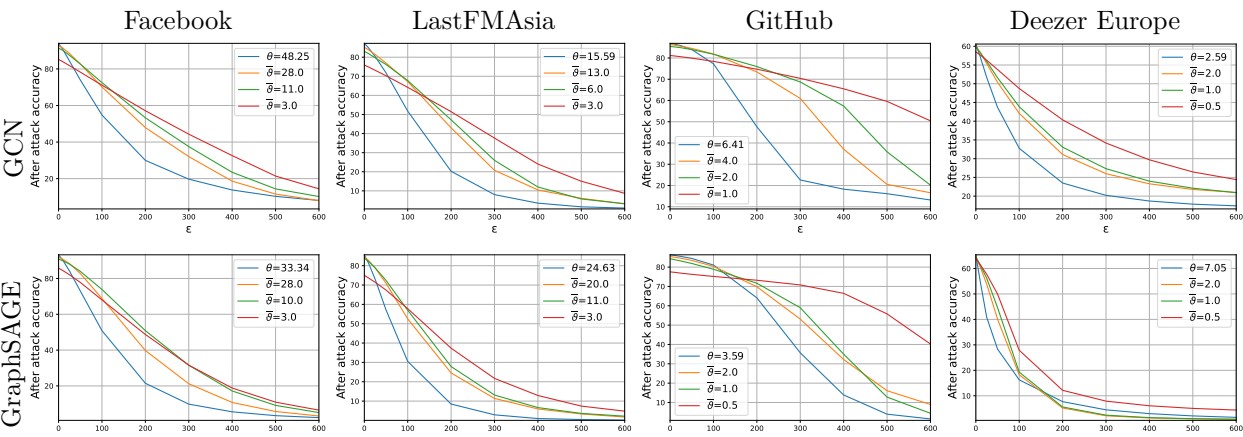

Figure 1: Accuracy - robustness tradeoff. Accuracy vs. $\ell_2$ norm of the test set perturbation ($\epsilon$). APGD attack with DL loss. The blue line represents the baseline model; the remaining lines depict constrained models with varying Lipschitz values $\overline{\vartheta}$. Top row: GCN model, bottom row: GraphSAGE model.

Descent (PGD) technique (Madry et al., 2018), and its parameter-free extension, Auto-PGD (APGD) (Croce & Hein, 2020), to evaluate the robustness of the models. Since we attack all the test nodes at once, we sum the loss over the attacked nodes. More information and details about the attackers can be found in Appendix C.3. We consider the task of transductive node classification, and the two architectures, GCN and GraphSAGE with the configuration described above. Our analysis involves examining how the baseline model and the models trained subject to spectral norm constraints behave when facing different levels of perturbations $\boldsymbol{\delta}$, evaluated with respect to the $\ell_2$ norm of the entire test set. We select the range of perturbation magnitudes $\|\boldsymbol{\delta}\|_2$ starting from the point where the models exhibit a minimal loss in performance, up to the point where the accuracy drops to (almost) zero, or it plateaus. In evaluating a total of six attack methods, formed by combining two attack techniques (PGD and APGD) with three distinct loss functions (Cross-Entropy (CE), Difference of Logits (DL), and Difference of Logits Ratio (DLR) (Carlini & Wagner, 2017; Croce & Hein, 2020)), we consistently found that the APGD attack outperformed PGD irrespective of the chosen loss function. Furthermore, regardless of the attack method, the DL loss demonstrated the strongest adversarial impact. Specifically, the APGD attack coupled with the DL loss emerged as the most powerful among the six attacks, across all datasets. Given this observation, we further compared the efficiency of the models subjected to an APGD-DL attack. A detailed discussion can be found in Appendix D.1.

Figure 2 displays the robustness of the models on the different datasets. Our models consistently outperform the baseline by a significant margin, showcasing robustness against adversarial attacks with minimal performance loss. These findings confirm that the theoretical concepts introduced in this paper lead to GNN models that are more robust to adversarial perturbations. Our models trained under spectral norm constraints perform much better than adversarially trained models across all datasets and configurations. AT models perform at most only slightly better than the conventionally trained models. Comparatively, the randomized smoothing approach exhibits good robustness in most cases when compared to the baseline model, but it is generally not as effective as our approach. It is worth noting that, although RS was proven effective, it requires multiple inferences (1000) to achieve good robustness, which is very time-consuming.

Although the considered Lipschitz constant was computed in the sense of $\ell_2$ norm, we tested our approach with an $\ell_\infty$ (APGD-DL) attack and observed similar or better performances across all datasets. These results are presented in Appendix D.2. Furthermore, to demonstrate the generalizability of our findings across a broad range of component-wise activation functions, we investigated four additional ones, besides ReLU, *i.e.*, sigmoid, hyperbolic tangent (tanh), leaky ReLU, and sigmoid linear unit (SiLU). We have found that our models consistently surpassed the baseline in these instances as well, highlighting their robustness against adversarial attacks. The results can be found in Appendix D.3.

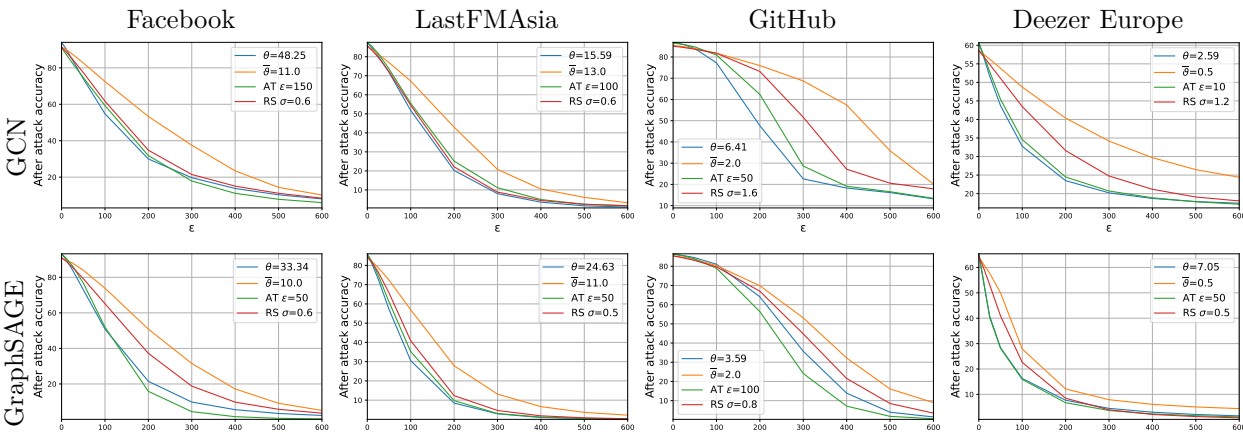

Figure 2: Robust training. Accuracy vs. $\ell_2$ norm of the test set perturbation ($\epsilon$). APGD attack with DL loss. The blue line represents the baseline model, the orange line represents our constrained model, the green line corresponds to AT, and the red line corresponds to RS. Top row: GCN model, bottom row: GraphSAGE model.

Finally, in the context of GCN networks, we conducted further comparisons to evaluate the effectiveness of our method. These comparisons entailed evaluating our method against Spectral Normalization (SN), considering both efficiency and performance. Additionally, we benchmarked our approach against three methods specifically designed to address perturbations to the graph structure: GCN-Jaccard (Wu et al., 2019), SVD-GCN (Entezari et al., 2020), and RGCN (Zhu et al., 2019). In the case of SN, our experiments revealed its ability to exhibit robustness against adversarial attacks compared to the baseline model. However, it fell short of delivering the performance achieved by our proposed approach (Figure 3a). On the other side, SN exhibits slightly faster training times compared to our method (4.9 ms per epoch, on average). The timing results are presented in Table 8, in Appendix D.4. The effectiveness of the methods designed to handle modifications to the graph structure is demonstrated in Figure 3b. Among these methods, SVD-GCN proves effective for large perturbations, but, for low to medium perturbations, it experiences a significant drop in accuracy. This behavior is observed across all datasets except Deezer Europe, where it demonstrates minimal robustness compared to the baseline model. Similarly, RGCN struggles to achieve high accuracy in the case of multiclass datasets. In contrast, GCN-Jaccard demonstrates robustness on multiclass datasets, although it shows limited effectiveness for binary datasets.

### 4.3 Memory and computational efficiency

Using existing techniques for controlling the Lipschitz constant of standard neural networks (often fully connected ones in the literature) (Cisse et al., 2017; Anil et al., 2019; Fazlyab et al., 2019; Huang et al., 2021; Neacşu et al., 2021; Araujo et al., 2022; Muthukumar & Sulam, 2023) is possible based on Equation (5). However, it is computationally inefficient and sometimes infeasible. The memory requirements associated with handling matrices of size $KN_i \times KN_{i-1}$ pose significant challenges, even for moderately sized graphs. Additionally, performing spectral normalization and, at the same time, imposing sparse algebraic constraints on these matrices proves to be computationally expensive.

To better exemplify the great reduction in memory allowed by our method, we consider the GCN used in this work and the two formulations: the fully connected one and ours. Table 3 provides the total number of elements in the weight matrices for each dataset. Assuming that each element is represented as a 32-bit floating-point number, we can estimate the memory occupied by the full matrices. For the specified datasets, the projected memory demands for these matrices range from 0.5 to 11.85 terabytes, surpassing the practical storage and handling capabilities of current GPUs. In contrast, our formulation introduces minimal additional memory overhead compared to training a conventional GCN, with the maximum memory requirement for the largest dataset being only a few dozen kilobytes.

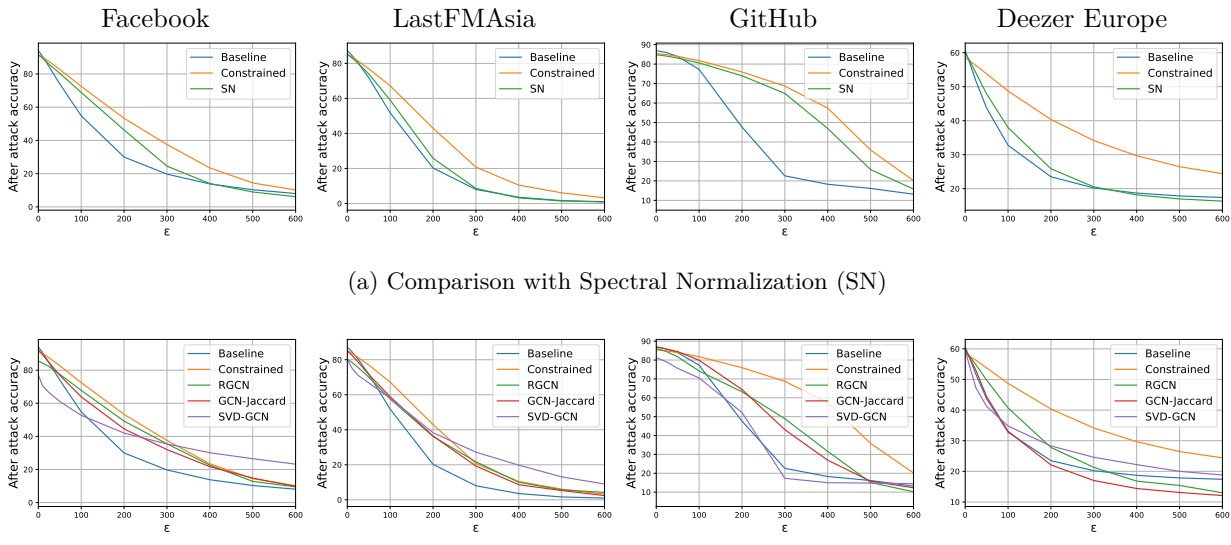

(a) Comparison with Spectral Normalization (SN)

(b) Comparison with methods designed against attacks on graph structure

Figure 3: Robust training for a GCN model. Accuracy vs. $\ell_2$ norm of the test set perturbation ($\epsilon$). APGD attack with DL loss.

Table 3: Comparison of matrix sizes between the fully connected and proposed formulations for GCN on various datasets.

| Dataset | Fully connected formulation | | | | Our formulation | | | |
|---|---|---|---|---|---|---|---|---|
| | $W_i^{(0)}$ | $W_i^{(1)}$ | $W_i^{(2)}$ | Total | $W_i^{(0)}$ | $W_i^{(1)}$ | $W_i^{(2)}$ | Total |
| Facebook | $1.0{\times}10^{12}$ | $1.3{\times}10^{11}$ | $3.2{\times}10^{10}$ | $1.2{\times}10^{12}$ | $2.0{\times}10^{3}$ | $2.6{\times}10^{2}$ | $6.4{\times}10^{1}$ | $2.4{\times}10^{3}$ |
| LastFM Asia | $1.2{\times}10^{11}$ | $1.5{\times}10^{10}$ | $1.7{\times}10^{10}$ | $1.5{\times}10^{11}$ | $2.0{\times}10^{3}$ | $2.6{\times}10^{2}$ | $2.9{\times}10^{2}$ | $2.6{\times}10^{3}$ |
| GitHub | $2.9{\times}10^{12}$ | $3.6{\times}10^{11}$ | $4.5{\times}10^{10}$ | $3.3{\times}10^{12}$ | $2.0{\times}10^{3}$ | $2.6{\times}10^{2}$ | $3.2{\times}10^{1}$ | $2.3{\times}10^{3}$ |
| Deezer Europe | $1.6{\times}10^{12}$ | $2.1{\times}10^{11}$ | $2.6{\times}10^{10}$ | $1.9{\times}10^{12}$ | $2.0{\times}10^{3}$ | $2.6{\times}10^{2}$ | $3.2{\times}10^{1}$ | $2.3{\times}10^{3}$ |

While our method incurs a slightly longer training duration (on average 1.36 times slower, per epoch) than baseline training, the overhead remains reasonable given the achieved robustness against adversarial attacks. The increased training times are more pronounced for larger and more complex datasets, but for smaller datasets, the difference in training time is minimal. In absolute values, our experiments show that the total time required to train a whole model was between $11.57s$ (simplest configuration) and $45.46s$ (biggest dataset), as opposed to $9.96s$ and $21.12s$, respectively, for the conventionally trained model. Thus, the additional training time can be considered quite acceptable. Notably, our method exhibits faster training times on average compared to AT, and RS does not require any additional training time. Despite its advantages, RS exhibits a major drawback with significantly slower inference times, ranging from 150 to 400 times slower in our experiments compared to the other methods. This may render RS impractical for some real-life applications, as it may take several seconds for a single inference. In contrast, our method does not introduce any overhead at inference. Comprehensive details on the time requirements for the experiments conducted in this study can be found in Appendix D.4.

## 5 Conclusion

This paper proposes a new approach for building GNNs subject to a Lipschitz bound by focusing on graphs with a constant topology and continuous features at each node. We derived Theorem 3.1 which provides a tight and tractable expression for this bound by assuming that the network weights are non-negative and an

additional mild symmetry property is met. Unlike the standard bound in Equation (9), the proposed bound is not layerwise separable. We showed that this bound can be further simplified for GraphSAGE and GCN networks. We validated our approach on four datasets for binary and multi-class classification problems. We showed that we can indeed control the Lipschitz constant of the GNN while maintaining good accuracy and benefiting from improved generalization properties. We also adapted the PGD and APGD attackers for generating white box adversarial attacks on our graph models. Overall, we observed better robustness than the one obtained by both adversarial training and randomized smoothing and, in the case of GCN networks, our method outperformed spectral normalization and methods designed against graph structure attacks.

## 6 Limitations and Future Work

Our work is subject to several limitations that should be taken into account when interpreting the results. We identify four main limitations:

(i) Non-negative Network Weights: Our proposed bound assumes that the network weights are non-negative. While this constraint did not in itself cause a significant drop in performance for our considered datasets, different conclusions might be observed on other datasets.

(ii) Undirected Graph: The graphs used in our experiments are undirected, which is the case for many datasets.

(iii) Graph Topology Attacks: Our work does not address graph topology attacks; in turn, we address feature attacks, which are seldom considered in the literature of GNNs.

(iv) Continuous Features and Untargeted Attacks: Our tests are based on continuous features, on which we perform untargeted attacks by PGD and APGD. However, other types of features or attacks could be considered.

Addressing these limitations could further enhance the validity and applicability of our findings. In future work, we intend to explore these constraints. Furthermore, we aim to explore the feasibility of employing norms other than the Euclidean one.

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

# A  Properties for Symmetric Matrices

In this section, we will consider the case when $M$ is symmetric with non-negative elements, which covers several of the examples mentioned in Section 3.1.

## A.1  Preliminary results

If $M$ is symmetric, its eigendecompositon is given by

$$M = UDU^\top, \tag{22}$$

where $U$ is an orthogonal matrix of $\mathbb{R}^{K \times K}$ and $D \in \mathbb{R}^{K \times K}$ is a diagonal matrix with diagonal elements

$$\lambda_1 \leq \cdots \leq \lambda_K. \tag{23}$$

Note that, since $M$ typically has zero diagonal terms, it is a so-called *hollow symmetric matrix*. This implies that its trace is zero, and it thus possesses both negative and positive eigenvalues.

**Lemma A.1.** *Suppose that $M$ satisfies the above conditions. Then*

$$\boldsymbol{W}^{(m)} \cdots \boldsymbol{W}^{(1)} = (U \otimes \mathrm{Id}_{N_m}) \Lambda (U^\top \otimes \mathrm{Id}_{N_0}), \tag{24}$$

*where $\Lambda \in \mathbb{R}^{(KN_m) \times (KN_0)}$ is a block diagonal matrix with $K$ diagonal blocks. Its $k$-th diagonal block is expressed as*

$$(w_0^{(m)} + \lambda_k w_1^{(m)}) \cdots (w_0^{(1)} + \lambda_k w_1^{(1)}) \in \mathbb{R}^{N_m \times N_0}. \tag{25}$$

*Proof.* By using the properties of Kronecker product, it follows from Equation (6) that, for every $i \in \{1, \dots, m\}$,

$$
\begin{aligned}
\boldsymbol{W}^{(i)} &= (UU^\top) \otimes w_0^{(i)} + (UDU^\top) \otimes w_1^{(i)} \\
&= (U \otimes \mathrm{Id}_{N_i})(\mathrm{Id}_K \otimes w_0^{(i)})(U^\top \otimes I_{N_{i-1}}) + (U \otimes \mathrm{Id}_{N_i})(D \otimes w_1^{(i)})(U^\top \otimes I_{N_{i-1}}) \\
&= (U \otimes \mathrm{Id}_{N_i})(\mathrm{Id}_K \otimes w_0^{(i)} + D \otimes w_1^{(i)})(U^\top \otimes I_{N_{i-1}}).
\end{aligned}
\tag{26}
$$

In addition, it can be noticed that

$$
\begin{aligned}
(U \otimes \mathrm{Id}_{N_i})(U^\top \otimes \mathrm{Id}_{N_i}) &= (UU^\top) \otimes (\mathrm{Id}_{N_i}\mathrm{Id}_{N_i}) \\
&= \mathrm{Id}_K \otimes \mathrm{Id}_{N_i} \\
&= \mathrm{Id}_{KN_i}.
\end{aligned}
\tag{27}
$$

Using Equation (26) and Equation (27) recursively yields

$$\boldsymbol{W}^{(m)} \cdots \boldsymbol{W}^{(1)} = (U \otimes \mathrm{Id}_{N_m}) \underbrace{(\mathrm{Id}_K \otimes w_0^{(m)} + D \otimes w_1^{(m)}) \cdots (\mathrm{Id}_K \otimes w_0^{(1)} + D \otimes w_1^{(1)})}_{\Lambda}(U^\top \otimes \mathrm{Id}_{N_0}). \tag{28}$$

For every $i \in \{1, \dots, m\}$, $\mathrm{Id}_K \otimes w_0^{(i)} + D \otimes w_1^{(i)}$ is a block diagonal matrix of size $(KN_i) \times (KN_{i-1})$ having $K$ block diagonal elements $(w_0^{(i)} + \lambda_k w_1^{(i)})_{1 \leq k \leq K}$. Thus $\Lambda$ is a block diagonal matrix with $K$ block diagonal elements of the form (25). $\square$

**Lemma A.2.** *Assume that, for every $i \in \{1, \dots, m\}$, $w_0^{(i)} \geq 0$ and $w_1^{(i)} \geq 0$. Let $\varphi$ be defined by Equation (10). If $(\mu, \overline{\mu}) \in \mathbb{R}^2$ are such that*

$$|\mu| \leq \overline{\mu}, \tag{29}$$

*then $\varphi(\mu) \leq \varphi(\overline{\mu})$. In particular, $\varphi$ is an increasing function on $[0, +\infty[$.*

*Proof.* We can write

$$(w_0^{(m)} + \mu w_1^{(m)}) \cdots (w_0^{(1)} + \mu w_1^{(1)}) = \sum_{i=0}^{m} \mu^i \widetilde{w}_i, \tag{30}$$

where $(\widetilde{w}_i)_{1 \le i \le m}$ are matrices in $[0, +\infty[^{N_m \times N_0}$. For every $x \in \mathbb{R}^{N_0}$,

$$\begin{aligned}
\|(w_0^{(m)} + \mu w_1^{(m)}) \cdots (w_0^{(1)} + \mu w_1^{(1)})x\|^2 &= \sum_{i=0}^{m} \sum_{j=0}^{m} \mu^{i+j} (\widetilde{w}_i x)^\top (\widetilde{w}_j x) \\
&\le \Big| \sum_{i=0}^{m} \sum_{j=0}^{m} \mu^{i+j} (\widetilde{w}_i x)^\top (\widetilde{w}_j x) \Big| \\
&\le \sum_{i=0}^{m} \sum_{j=0}^{m} |\mu|^{i+j} (\widetilde{w}_i |x|)^\top (\widetilde{w}_j |x|) \\
&\le \sum_{i=0}^{m} \sum_{j=0}^{m} \overline{\mu}^{i+j} (\widetilde{w}_i |x|)^\top (\widetilde{w}_j |x|),
\end{aligned} \tag{31}$$

where $|x|$ denotes the vector whose components are the absolute values of those of $x$ and we have used in the last inequality the nonnegativity of the elements of matrices $(\widetilde{w}_i)_{1 \le i \le m}$. We deduce from Equation (31) that

$$\|(w_0^{(m)} + \mu w_1^{(m)}) \cdots (w_0^{(1)} + \mu w_1^{(1)})x\| \le \|(w_0^{(m)} + \overline{\mu} w_1^{(m)}) \cdots (w_0^{(1)} + \overline{\mu} w_1^{(1)})|x|\|. \tag{32}$$

Consequently,

$$\begin{aligned}
\varphi(\mu) &= \sup_{x \in \mathbb{R}^{N_0} \setminus \{0\}} \frac{\|(w_0^{(m)} + \mu w_1^{(m)}) \cdots (w_0^{(1)} + \mu w_1^{(1)})x\|}{\|x\|} \\
&\le \sup_{x \in \mathbb{R}^{N_0} \setminus \{0\}} \frac{\|(w_0^{(m)} + \overline{\mu} w_1^{(m)}) \cdots (w_0^{(1)} + \overline{\mu} w_1^{(1)})|x|\|}{\||x|\|} \\
&= \sup_{a \in [0,+\infty[^{N_0} \setminus \{0\}} \frac{\|(w_0^{(m)} + \overline{\mu} w_1^{(m)}) \cdots (w_0^{(1)} + \overline{\mu} w_1^{(1)})a\|}{\|a\|} \\
&\le \sup_{x \in \mathbb{R}^{N_0} \setminus \{0\}} \frac{\|(w_0^{(m)} + \overline{\mu} w_1^{(m)}) \cdots (w_0^{(1)} + \overline{\mu} w_1^{(1)})x\|}{\|x\|} \\
&= \varphi(\overline{\mu}).
\end{aligned} \tag{33}$$

$\square$

## A.2 Proof of Theorem 3.1

In Scaman & Virmaux (2018) and Combettes & Pesquet (2020), an upper bound on the Lipschitz constant of a feed forward network with weight $(\boldsymbol{W}^{(i)})_{1 \le i \le m}$ is derived for a broad class of activation functions (see Proposition 5.5 in Combettes & Pesquet (2020)). This bound is tight in the sense that it is finer than most of the existing bounds, in particular the classical separable one in Equation (9). This can also be checked numerically on simulated networks when compared with other recent methods for computing Lipschitz bounds (Fazlyab et al., 2019). The limitation of this upper bound is that it is NP-hard to compute (Scaman & Virmaux, 2018), but it can however be lower bounded by the more tractable expression

$$\vartheta = \|\boldsymbol{W}^{(m)} \cdots \boldsymbol{W}^{(1)}\|_{\mathrm{S}}. \tag{34}$$

This corresponds to the optimal Lipschitz constant of a linear neural network, where activations reduce to the identity function. When the network has non-expansive activation operators and non-negative weights, it is shown in Proposition 5.10 of Combettes & Pesquet (2020) that this lower bound $\vartheta$ and the tight

expression of the Lipschitz constant coincide. Note that Equation (34) is only proved to be an upper bound when using activation functions that apply component-wise (except for the last layer). According to the Perron-Frobenius theorem[3], if $\boldsymbol{W}^{(m)} \cdots \boldsymbol{W}^{(1)}$ has non-negative elements, then its spectral norm is an eigenvalue of this matrix which admits an associated eigenvector $\boldsymbol{e}$ with non-negative components. If this vector is used as the input of a feed forward neural network with non-negative weights $(\boldsymbol{W}^{(i)})_{1 \leq i \leq m}$, no bias, and ReLU activations at all layers, it is straightforward to show that the output of the neural network is $\boldsymbol{W}^{(m)} \cdots \boldsymbol{W}^{(1)} \boldsymbol{e} = \|\boldsymbol{W}^{(m)} \cdots \boldsymbol{W}^{(1)}\|_{\mathrm{S}} \boldsymbol{e}$. The bound in Equation (34) being thus attained in this case, it is optimal. However, it is noteworthy that $\vartheta$ is an upper bound on the Lipschitz constant of non-negative neural networks for other activation functions than ReLU. In addition, this result holds for inputs with arbitrary signs.

Using the same notation as in Appendix A.1, it follows from Lemma A.1 and the orthogonality of matrices $U \otimes \mathrm{Id}_{N_m}$ and $U^\top \otimes \mathrm{Id}_{N_0}$ that

$$\|\boldsymbol{W}^{(m)} \cdots \boldsymbol{W}^{(1)}\|_{\mathrm{S}} = \|\Lambda\|_{\mathrm{S}}. \tag{35}$$

Because of the block diagonal structure of $\Lambda$ and the definition of $\varphi$ in Equation (10),

$$\|\boldsymbol{W}^{(m)} \cdots \boldsymbol{W}^{(1)}\|_{\mathrm{S}} = \max_{1 \leq k \leq K} \varphi(\lambda_k). \tag{36}$$

Since $M$ is a matrix with non-negative elements, it follows from the Perron-Frobenius theorem that

$$(\forall k \in \{1, \ldots, K-1\}) \qquad |\lambda_k| \leq \lambda_K. \tag{37}$$

We deduce from Lemma A.2 that

$$(\forall k \in \{1, \ldots, K-1\}) \qquad \varphi(\lambda_k) \leq \varphi(\lambda_K). \tag{38}$$

By using Equation (36), this yields Equation (11). When the considered GNN has no bias and uses only ReLU activations, the optimality of the bound in Equation (11) follows from our previous comments regarding the optimality of (34).

### A.3 Remarks on the use of Theorem 3.1

(i) For every node $v$, let

$$d_v = \sum_{u \in \mathcal{N}(v)} M_{v,u}. \tag{39}$$

From standard results in spectral graph theory,[4] the following inequality is satisfied

$$\frac{\sum_{v \in V} d_v}{K} \leq \lambda_K \leq \max_{v \in V} d_v. \tag{40}$$

It thus follows from Lemma A.2 that under the assumptions of Theorem 3.1,

$$\varphi\Big(\frac{\sum_{v \in V} d_v}{K}\Big) \leq \|\boldsymbol{W}^{(m)} \cdots \boldsymbol{W}^{(1)}\|_{\mathrm{S}} \leq \varphi(\max_{v \in V} d_v). \tag{41}$$

(ii) Assume that $M$ corresponds to the normalized (possibly weighted) adjacency matrix of an undirected graph. Then $M = (D^{\boldsymbol{\varepsilon}})^{-1/2} A^{\boldsymbol{\varepsilon}} (D^{\boldsymbol{\varepsilon}})^{-1/2}$ where $A^{\boldsymbol{\varepsilon}}$ is the non-negative-valued symmetric matrix defined in Equation (2) and $D^{\boldsymbol{\varepsilon}}$ is the diagonal matrix with diagonal terms $\mathrm{d}^{\boldsymbol{\varepsilon}}(v) = \sum_{u \in \mathcal{N}(v)} \varepsilon_{v,u}$, $v \in V$. The case of an unweighted normalized adjacency matrix is recovered when, for every $v \in V$ and $u \in \mathcal{N}(v)$, $\varepsilon_{v,u} = 1$. Using graph spectral theory concepts, it can be shown that the maximum eigenvalue of $M$ is $\lambda_K = 1$.

---

[3]Carl D. Meyer, Matrix Analysis and Applied Linear Algebra, SIAM, 2000.
[4]Daniel A. Spielman, Spectral and Algebraic Graph Theory, Yale University, 2019.

*Proof.* Although this result is known in the case of unweighted adjacency matrices, we provide a proof for completeness. For every $(v, u) \in V^2$, set

$$
\bar{\varepsilon}_{v,u} = \begin{cases} \dfrac{\varepsilon_{v,u}}{\mathrm{d}^{\boldsymbol{\varepsilon}}(v)} & \text{if } u \in \mathcal{N}(v) \\ 0 & \text{otherwise.} \end{cases}
\tag{42}
$$

By symmetry, we have, for every $x = (x_v)_{v \in V} \in \mathbb{R}^K$,

$$
\sum_{(v,u) \in V^2} (\sqrt{\bar{\varepsilon}_{v,u}} x_v - \sqrt{\bar{\varepsilon}_{u,v}} x_u)^2
$$

$$
= \sum_{v \in V} \sum_{u \in \mathcal{N}(v)} \bar{\varepsilon}_{v,u} x_v^2 + \sum_{u \in V} \sum_{v \in \mathcal{N}(u)} \bar{\varepsilon}_{u,v} x_u^2 - 2 \sum_{(v,u) \in V^2} \sqrt{\bar{\varepsilon}_{v,u} \bar{\varepsilon}_{u,v}} x_v x_u
$$

$$
= 2 \left( \sum_{v \in V} x_v^2 - \sum_{v \in V, u \in \mathcal{N}(v)} \frac{\varepsilon_{v,u}}{\sqrt{\mathrm{d}^{\boldsymbol{\varepsilon}}(v) \mathrm{d}^{\boldsymbol{\varepsilon}}(u)}} x_v x_u \right)
$$

$$
= 2 x^\top L^{\boldsymbol{\varepsilon}} x,
\tag{43}
$$

where $L^{\boldsymbol{\varepsilon}} = \mathrm{Id}_K - M$ is the weighted Laplacian matrix of the graph. Since we have proved that, for every $x \in \mathbb{R}^K$, $x^\top L^{\boldsymbol{\varepsilon}} x \geq 0$, $L^{\boldsymbol{\varepsilon}}$ is a positive semi-definite matrix. This means that

$$
\lambda_K \leq 1.
\tag{44}
$$

On the other hand, if $x = (D^{\boldsymbol{\varepsilon}})^{1/2} 1_K$, we have

$$
Mx = (D^{\boldsymbol{\varepsilon}})^{-1/2} A^{\boldsymbol{\varepsilon}} 1_K
$$

$$
= (D^{\boldsymbol{\varepsilon}})^{-1/2} \left( \sum_{u \in \mathcal{N}(v)} \varepsilon_{v,u} \right)_{v \in V} = x.
\tag{45}
$$

This show that $x$ is an eigenvector of $M$ associated with eigenvalue 1. Consequently, the upper bound is attained in Equation (44). $\square$

(iii) When the weights are not assumed to be non-negative, it is also possible to evaluate a Lipschitz constant of an individual layer under the symmetry assumption for $M$. Indeed, it follows from Equation (26), that

$$
\|\boldsymbol{W}^{(i)}\|_{\mathrm{S}} = \|\mathrm{Id}_K \otimes w_0^{(i)} + D \otimes w_1^{(i)}\|_{\mathrm{S}}
$$

$$
= \max_{1 \leq k \leq K} \|w_0^{(i)} + \lambda_k w_1^{(i)}\|_{\mathrm{S}}.
\tag{46}
$$

By using the triangle inequality, we have also the upper bound

$$
\|\boldsymbol{W}^{(i)}\|_{\mathrm{S}} \leq \|w_0^{(i)}\|_{\mathrm{S}} + \lambda_K \|w_1^{(i)}\|_{\mathrm{S}}.
\tag{47}
$$

In particular, if $w_1^{(i)} = 0$ then Equation (46) reduces to

$$
\|\boldsymbol{W}^{(i)}\|_{\mathrm{S}} = \|w_0^{(i)}\|_{\mathrm{S}}.
\tag{48}
$$

## A.4  GCN case

Suppose that, for every $i \in \{1, \dots, m\}$, $w_1^{(i)} \geq 0$. The main mathematical difference in the analysis of GCN is that matrix $\widetilde{M}$ has nonzero diagonal elements. However, it remains symmetric with non-negative elements which are the sufficient requirements in the proof of Theorem 3.1. The result thus extends straightforwardly. Let $\widetilde{\lambda}_K > 0$ be the maximum eigenvalue of $\widetilde{M}$. Then

$$
\|\boldsymbol{W}^{(m)} \cdots \boldsymbol{W}^{(1)}\|_{\mathrm{S}} = \widetilde{\lambda}_K^m \|w_1^{(m)} \cdots w_1^{(1)}\|_{\mathrm{S}}.
\tag{49}
$$

Similarly, it is easy to see that the reasoning in Appendix A.3 (ii) remains valid for matrix $\widetilde{M} = \widetilde{D}^{-1/2} \widetilde{A} \widetilde{D}^{-1/2}$, so allowing us to conclude that $\widetilde{\lambda}_K = 1$.

# B   Lipschitz Constant Estimates

As discussed in Section 3.2, the Lipschitz constant provides an upper bound on the ratio between the output and input variations for a given metric. An accurate estimation of this bound is mandatory to ensure a good trade-off between robustness and performance. Therefore, we aim to compute an estimate of the Lipschitz constant for GNNs that is as tight as possible.

In Table 4, we numerically exemplify the difference between the different formulations for estimating a Lipschitz constant of a network (Equations (9) and (34)). We consider the two particular GNNs discussed in Sections 3.3 and 3.4, *i.e.*, symmetrical GraphSAGE and GCN with non-negative weights. We train networks of varying depths, from shallow to deep, on the Facebook dataset and compute a Lipschitz constant estimates using the two formulas mentioned above. The hidden dimension of each layer is $N_i = 16$ for $i \in \{1, \ldots, m\}$. From the results presented in the table, it can be clearly seen that the more complex the network, the bigger the discrepancy between the loose and the tight bounds. The difference between the two is higher in the case of GraphSAGE than GCN, possibly due to the higher number of parameters of the former. Note that, from a theoretical viewpoint, the ratio between $\vartheta$ and $\theta$ could be arbitrarily small for some choices of the weights.

Table 4: Loose and tight Lipschitz constant estimates for networks of different depths trained on the Facebook dataset.

| Number of layers | GCN | | GraphSAGE | |
|---|---|---|---|---|
| | $\theta = \prod_{i=1}^{m} \|\boldsymbol{W}^{(i)}\|_{\mathrm{S}}$ | $\vartheta = \|\boldsymbol{W}^{(m)} \cdots \boldsymbol{W}^{(1)}\|_{\mathrm{S}}$ | $\theta = \prod_{i=1}^{m} \|\boldsymbol{W}^{(i)}\|_{\mathrm{S}}$ | $\vartheta = \|\boldsymbol{W}^{(m)} \cdots \boldsymbol{W}^{(1)}\|_{\mathrm{S}}$ |
| 1 | 226.04 | 207.29 | 170.19 | 148.48 |
| 2 | 375.06 | 345.6 | 880.2 | 626.62 |
| 3 | 564.84 | 471.6 | 833.07 | 589.31 |
| 4 | 578.85 | 406.3 | 904.22 | 425.66 |
| 5 | 294.41 | 206.66 | 1034.19 | 349.39 |
| 10 | 1819.86 | 567.55 | 7260.37 | 1261.41 |

# C   Comprehensive Experimental Description

## C.1   Dataset details

We considered various publicly available, real-world social networks and web graph datasets:

- **Facebook Page-Page** (Rozemberczki et al., 2021): Nodes represent verified Facebook pages and edges mutual likes. The features are extracted from the description of the page, and the label is the page category.

- **GitHub** (Rozemberczki et al., 2021): Nodes are developers on GitHub and edges represent mutual follower relationships. Node features are information about the repository and the owner (*e.g.*, starred repositories, location, etc). The task is to classify nodes as machine learning or web developers.

- **Deezer Europe** (Rozemberczki & Sarkar, 2020): Nodes represent users and edges are friendships. The binary classification task is the prediction of the users' gender, considering the liked artists as features.

- **LastFM Asia** (Rozemberczki & Sarkar, 2020): Nodes represent Asian users, edges are friendships and features consist of artists liked by the users. The task is to predict the nationality of the user.

Their descriptive statistics are presented in Table 5.

Table 5: Statistics of the graph datasets used for the evaluation of node classification algorithms

| Dataset | Nodes | Edges | Features per node | Classes |
|---|---|---|---|---|
| Facebook | 22470 | 342004 | 128 | 4 |
| GitHub | 37300 | 578006 | 128 | 2 |
| LastFM Asia | 7624 | 55612 | 128 | 18 |
| Deezer Europe | 28281 | 185504 | 128 | 2 |

## C.2 Experimental configuration

In our experiments, we trained each model for 2000 iterations, following a strategy that reduces the learning rate by a factor of 10 if the performance of the model did not improve for 100 epochs. We trained the model using the Adam optimizer, in the standard configuration ($\beta_1 = 0.9$, $\beta_2 = 0.99$) with $\ell_2$ penalty. The weight decay was tuned using 5-fold cross-validation for each dataset and GNN formulation. All the layers have a hidden dimension of $N_i = 16$. Moreover, we have implemented an early stopping mechanism, with a patience of 200 iterations. The DBF algorithm uses a maximum number of 100 iterations, but also employs an *early stopping* mechanism. At each step of the algorithm, we compute the current Lipschitz constant and compare it to the target Lipschitz constant. If the difference between the two is lower than a tolerance parameter (chosen as 0.01), we stop the algorithm. Typically, the algorithm requires more iterations during the first few epochs and then just a few for subsequent ones. The common parameters for all datasets and network architectures are summarized in Table 6, while the particular weight decay values are presented in Table 7.

Table 6: Training hyperparameters

| | Parameter | Value |
|---|---|---|
| Training | Hidden dimension | 16 |
| | Max num. epochs | 2000 |
| | Early stopping patience | 200 |
| | Initial learning rate | 0.01 |
| | Learning rate patience | 100 |
| DFB algorithm | Max num. iterations | 100 |
| | $\alpha$ | 2.1 |
| | Max. Lipschitz difference | 0.01 |

For adversarial training (AT), the adversarial samples were generated using the Auto-PGD attack (Croce & Hein, 2020) with a Difference of Logits (DL) loss, for different perturbation levels *i.e.*, $\varepsilon \in \{10, 50, 100, 150, 200, 300, 400, 500, 600, 800, 900, 1000\}$, which represent the $\ell_2$ norm of the perturbation on the entire test set.

For randomized smoothing (RS), we adapted the method presented in Cohen et al. (2019) to the graph domain. We trained the base classifier (GCN or GraphSAGE) with inputs perturbed by Gaussian noise. Then, during inference, we ran noise-corrupted copies of the input through the base classifier and returned the class that appeared most frequently. We used the hypothesis test from Hung & Fithian (2019) to calibrate the abstention threshold to bound by $\alpha$ the probability of returning an incorrect answer. Because of the high time requirements of this method, we chose $\delta = 10^{-5}$ so that we get a good trade-off between prediction time and failure probability. Thus, we chose $N$ such that our prediction function with failure probability $\alpha = 0.001$ abstained at most 10.5% of the time for all the datasets, architectures, and values of $\sigma$. We chose $N = 1000$ yielding an abstention rate of at most 1.3% for $\sigma = 0.1$ , 2.6% for $\sigma = 0.25$ , 3.2% for $\sigma = 0.5$ , 6.4% for $\sigma = 0.75$ , 8.7% for $\sigma = 1$ and 10.4% for $\sigma = 2$. We performed the experiments for multiple values of the noise level $\sigma$, *i.e.*, $\sigma \in \{0.1, 0.25, 0.5, 0.6, 0.7, 0.8, 0.9, 1, 1.2, 1.4, 1.6, 1.8, 2\}$. The Gaussian noise was applied element-wise to each feature vector of the input data.

Table 7: L2 penalty for each dataset and network architecture

| Network | Dataset | L2 penalty |
|---------|---------|------------|
| GCN | Facebook | 0.0005 |
| | GitHub LastFM Asia Deezer Europe | 0.005 |
| GraphSAGE | Facebook | 0.005 |
| | GitHub | 0.05 |
| | LastFM Asia Deezer Europe | 0.01 |

In the context of GCN networks, we considered several additional methods, *i.e.*, Spectral Normalization (SN) (Miyato et al., 2018), GCN-Jaccard (Wu et al., 2019), SVD-GCN (Entezari et al., 2020), and RGCN (Zhu et al., 2019). For SN we trained models subject to various upper Lipschitz bound constraints *i.e.*, $\theta \in [1, 30]$. The implementations of the strategies aiming at mitigating the effects of perturbations on the graph structure (GCN-Jaccard, SVD-GCN, RGCN) were sourced from the Li et al. (2020) library. For each of these methods, we adhered to the default parameters as detailed in the respective papers. To find the best model for SVD-GCN, GCN-Jaccard, and RGCN we varied the number of singular values ($k \in \{5, 10, 15, 20, 25, 30, 40, 50\}$), the similarity threshold ($\tau \in \{0, 0.05, 0.1, 0.3, 0.5, 0.7\}$), and the hyperparameter in setting the attention weights ($\gamma \in \{0.1, 0.3, 0.5, 0.7, 1, 2, 5\}$), respectively. In the case of GCN-Jaccard, dissimilar edges were dropped based on the cosine similarity.

The same experimental configuration presented in Tables 6 and 7 is also applicable for AT, RS, SN, SVD-GCN, GCN-Jaccard, and RGCN.

### C.3 Attacks and robustness evaluation

We adapted the classical Projected Gradient Descent (PGD) (Madry et al., 2018) to evaluate the robustness of the models, building upon an existing implementation available on GitHub (Rony & Ben Ayed, 2023). Since we attack several nodes at once, we sum the loss over the attacked nodes. The PGD iterations on the perturbation $\boldsymbol{\delta} \in \mathbb{R}^{KN_0}$, for the input graph features $\boldsymbol{x}$ with labels $\boldsymbol{c} = (c_v)_{v \in V}$, read

$$\boldsymbol{g}^{(t)} = \nabla_{\boldsymbol{\delta}} \Big[ \sum_{v \in \mathcal{V}} \mathcal{L}((T(\boldsymbol{x} + \boldsymbol{\delta}^{(t)}))_v, c_v) \Big]$$

$$\boldsymbol{\delta}^{(t+1)} = \mathcal{P}_{\mathcal{B}_\varepsilon} \Big( \boldsymbol{\delta}^{(t)} + \eta \frac{\boldsymbol{g}^{(t)}}{\|\boldsymbol{g}^{(t)}\|_2} \Big)$$

(50)

where $\mathcal{V}$ is the subset of nodes to attack, $\mathcal{B}_\varepsilon = \{x \mid \|x\|_2 \le \varepsilon\}$ is the closed Euclidean ball of radius $\varepsilon$, $\eta$ is the step-size, and $\mathcal{L}$ is a classification loss. For the attack, we use a relative step-size of $1/30$, *i.e.*, $\eta = \frac{\varepsilon}{30}$. Additionally, it is known that PGD with Cross-Entropy (CE) can fail on robust models. One workaround is to use a loss that better aligns with the misclassification objective (for untargeted attacks): either the Difference of Logits (DL) (Carlini & Wagner, 2017), or the more recently proposed Difference of Logits Ratio (DLR) (Croce & Hein, 2020). Note that DLR can only be used for datasets with more than two classes because it relies on the top three (or four for targeted attacks) logits. Therefore, we perform attacks with CE and DL losses, *and DLR when possible*. Finally, we use a budget of 100 steps for the attack, with 9 additional random restarts, totalling 1 000 steps.

In addition, we also adapted the Auto-PGD attack (APGD) introduced by Croce & Hein (2020), whose only free parameter is the number of iterations. The key distinction in APGD, compared to the conventional

PGD, lies in the selection of the step size at each iteration. In APGD, the step size is dynamically adjusted based on the available budget and the optimization progress. Furthermore, when the step size is reduced, the maximization process restarts from the best point obtained up to that point. As for PGD, we perform attacks with CE and DL losses, *and DLR when possible*, for 1 000 steps.

## D  Additional Results

### D.1  Impact of the choice of the loss function on the performance of the attackers

We studied the impact of the loss function choice on the success of the PGD and APGD attacks in order to determine the strongest attack for all datasets and network configurations. Therefore, we considered three different losses for the attackers: Cross-Entropy (CE), Difference of Logits (DL) (Carlini & Wagner, 2017), and Difference of Logits Ratio (DLR) (Croce & Hein, 2020). Because DLR can only be used for datasets with more than two classes, we perform the PGD attack with CE and DL losses, and DLR when possible. Considering the task of transductive node classification, each model having $m = 3$ layers and $N_i = 16$ for $i \in \{1, 2\}$, we analysed how the baseline model and the one trained subjected to spectral constraints behave when facing different levels of perturbations $\boldsymbol{\delta}$, computed with respect to the $\ell_2$ norm. The constrained model (*i.e.*, choice of $\overline{\vartheta}$) was selected as the model with the largest area under the accuracy-robustness curve while ensuring that it exhibits a decrease in relative accuracy on the validation dataset of less than 3% when compared to the conventionally trained model. The results for the constrained models are depicted in Figure 4. In evaluating a total of six attack methods, formed by combining two attack techniques (PGD and APGD) with the three distinct loss functions (CE, DL, and DLR), we consistently observed that, regardless of the attack method, DL loss exhibited the strongest adversarial impact, while CE loss resulted in the least effective attack. Specifically, for the Facebook dataset, at lower levels of perturbation, the DLR loss led to a slightly larger decrease in performance. Interestingly, the APGD attack demonstrated greater strength than PGD in the case of CE and DLR losses. However, for the DL loss, the two attackers exhibited almost identical effectiveness.

### D.2  Adversarial robustness evaluation with respect to $\ell_\infty$ norm

While our method primarily focuses on computing a Lipschitz constant within the $\ell_2$ norm framework, it is worth noting that we evaluated the robustness of our models against an $\ell_\infty$ attack. We assessed the performance of models subjected to non-negativity and spectral norm constraints against those employing adversarial training (AT) and randomized smoothing (RS). We evaluated all models against the APGD-DL attack, considering various $\ell_\infty$ norms for perturbations on nodes in the test set, specifically $\|\boldsymbol{\delta}\|_\infty \in [0.01, 1]$. The accuracy curves are presented in Figure 5. We found that the constrained models behaved similarly or better against an $\ell_\infty$ attack as compared to an $\ell_2$ APGD-DL attack. Our models consistently outperform standard models across all datasets and configurations. AT models show, at most, a marginal improvement over conventionally trained models. Although RS exhibits better robustness compared to the baseline model, it generally falls short of the effectiveness demonstrated by our approach.

### D.3  Results on alternative activation functions

In the experiments conducted in this paper, we employed the rectified linear unit (ReLU) activation function for the intermediate layers, which is the standard choice for our explored models (Kipf & Welling, 2016; Hamilton et al., 2017a). To demonstrate the generalizability of our findings across a broad range of component-wise activation functions, we investigated four additional ones, namely, sigmoid, hyperbolic tangent (tanh), leaky ReLU, and sigmoid linear unit (SiLU). As for ReLU, we assessed the performance of the baseline and constrained models against the $\ell_2$ APGD-DL attack. Figure 6 presents the results for these models combined with the four activation functions, where *ct.* denotes our constrained models, and the baseline models are represented with a dashed line. In all cases, our models surpassed the baseline in these instances as well, highlighting that our findings hold for a diverse array of activation functions, extending beyond the conventional ReLU choice.

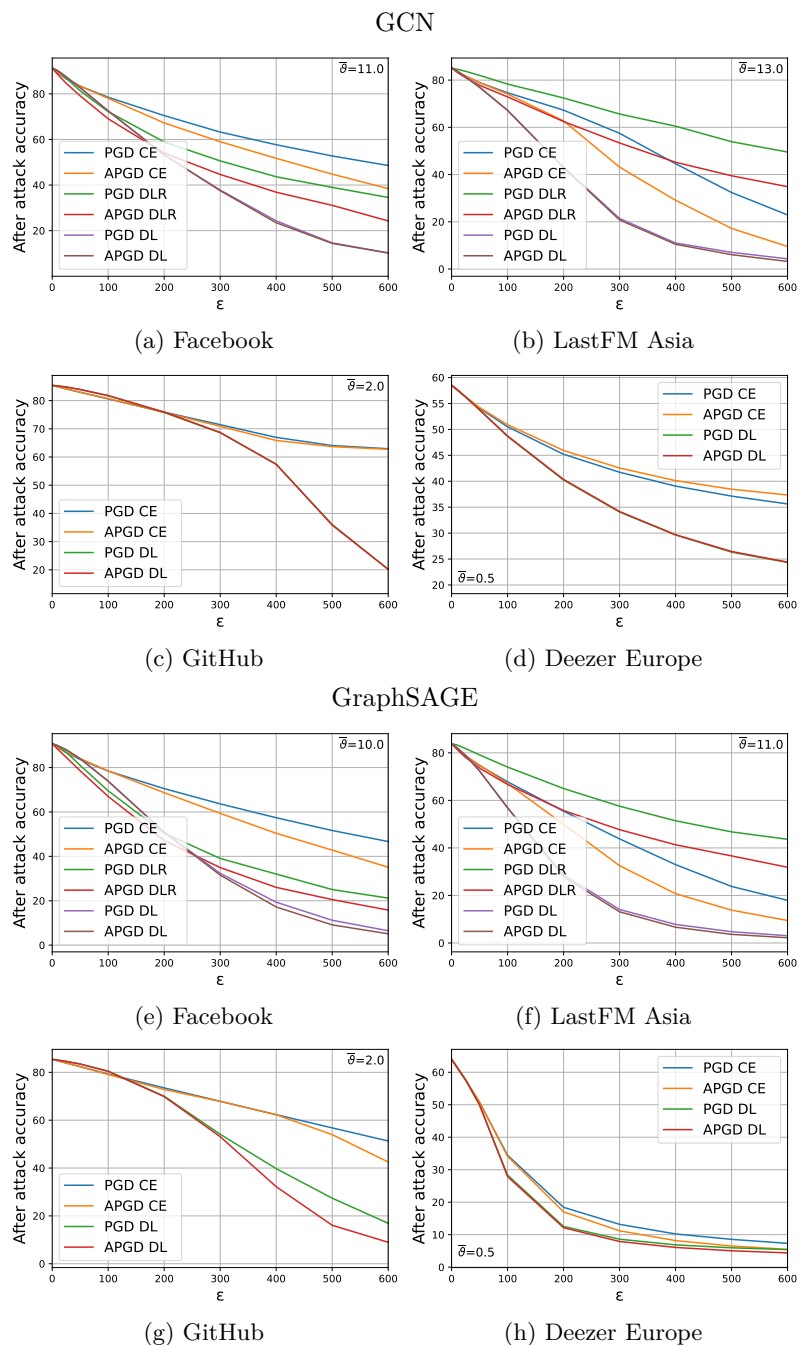

Figure 4: Impact of the loss function on the performance of the attackers. Accuracy vs. $\ell_2$ norm of the test set perturbation ($\epsilon$). PGD and APGD attack with different loss functions on constrained models. (a)-(d) GCN model. (e)-(h) GraphSAGE model.

## D.4  Time requirements and experimental durations

All experiments conducted in this paper were performed on an NVIDIA A100 80GB GPU. The cumulative execution time for all experiments amounted to approximately one week.

Tables 8 and 9 present the training times for the baseline model, the constrained model, adversarial training (AT), randomized smoothing (RS), and spectral normalization (SN), averaged across 10 splits. As discussed

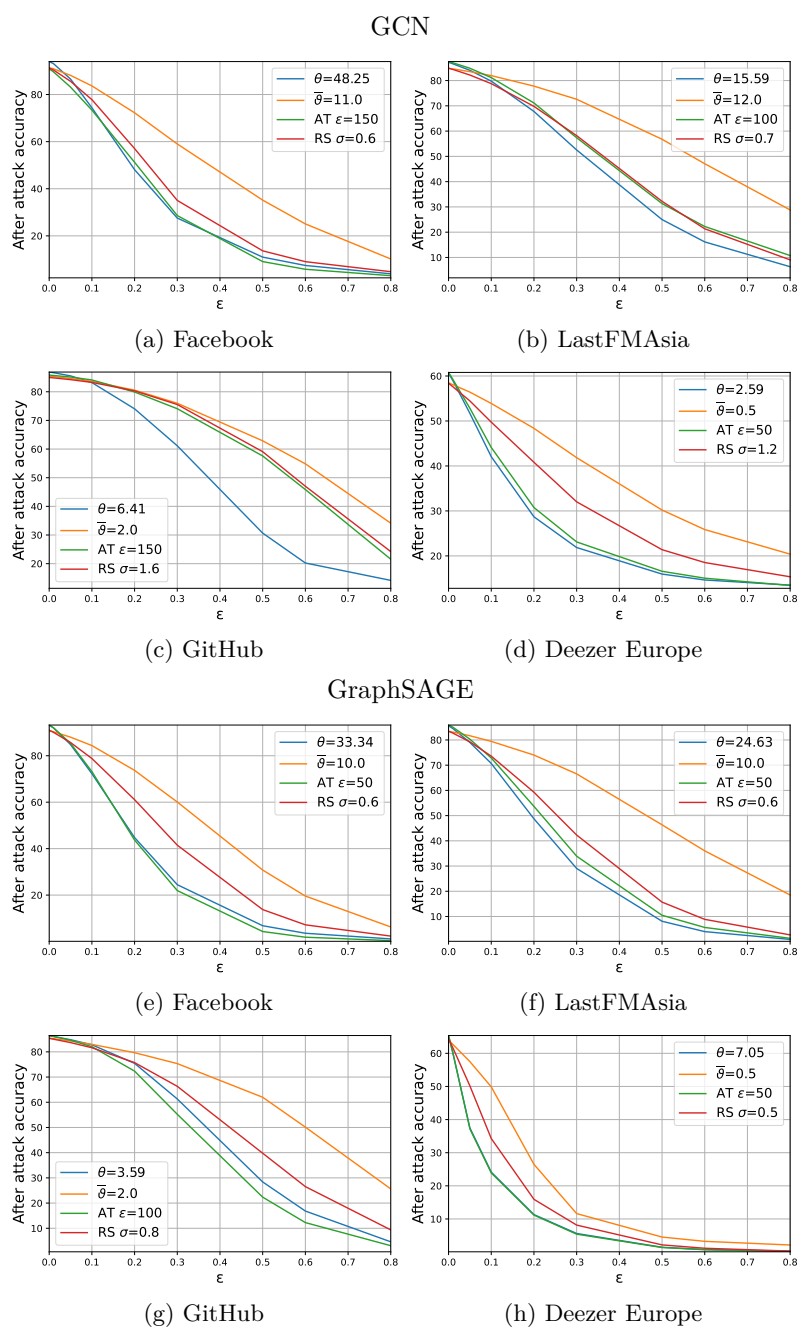

Figure 5: Robust training. Accuracy vs. $\ell_\infty$ norm of the test set perturbation ($\epsilon$). APGD attack with DL loss. The blue line represents the baseline model, the orange line represents the constrained model, the green line corresponds to AT, and the red line corresponds to RS. (a)-(d) GCN model. (e)-(h) GraphSAGE model.

in Appendix C.2, we employ an early stopping mechanism with a patience of 200 iterations. Table 8 presents the average time per epoch for training each network. Training the constrained GCN model takes, on average, $1.38 \times$ the time needed to train the baseline model per epoch. For the GraphSAGE architecture, this ratio is 1.35. Imposing the constraint takes a longer time for more complex datasets, such as Facebook and LastFM Asia. Naturally, the average time per epoch for RS and AT (without taking into account the additional time for generating the adversarial samples and pre-training) is comparable to the time needed to train the

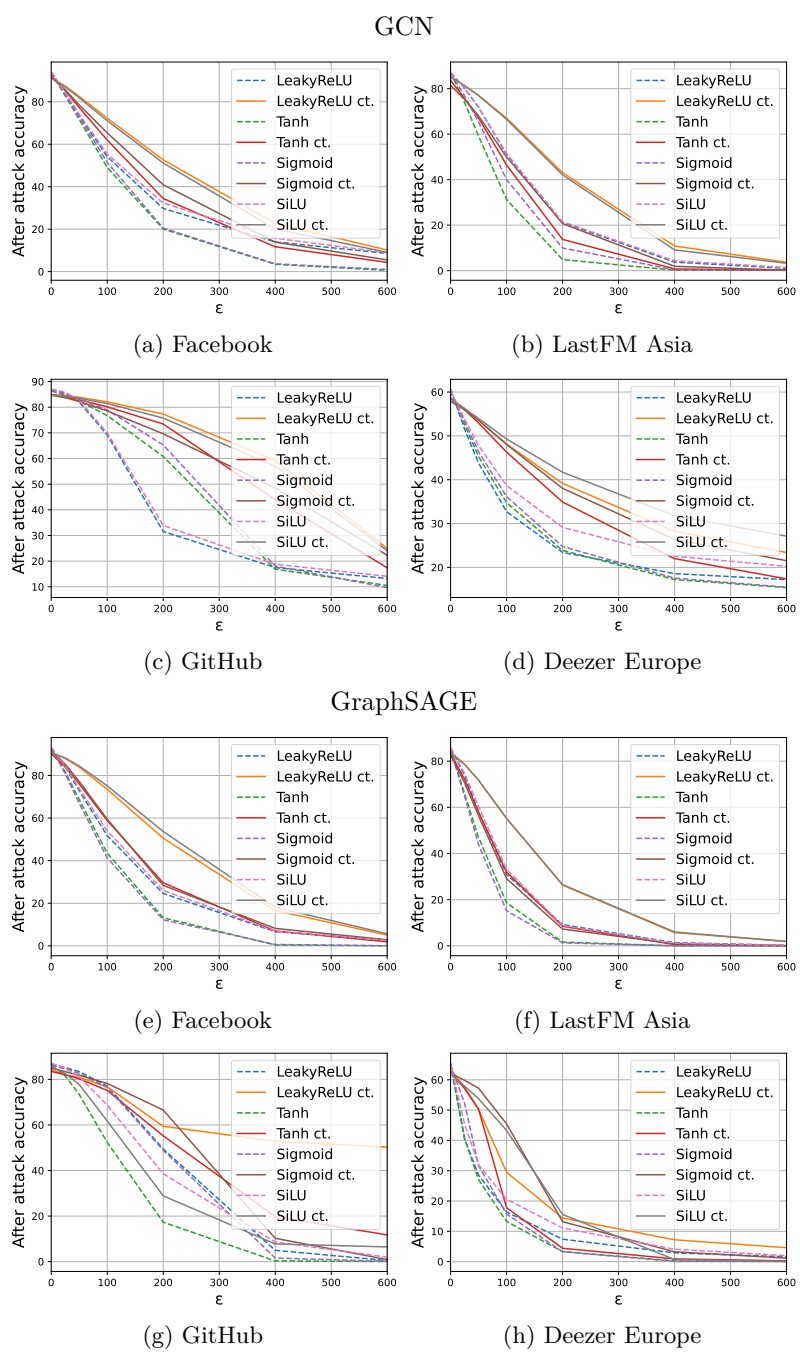

Figure 6: Results on alternative activation functions. Accuracy vs. $\ell_2$ norm of the test set perturbation ($\epsilon$). APGD attack with DL loss. The constrained models are plotted with a solid line and denoted by *ct.*, and the baseline models are represented with a dashed line. (a)-(d) GCN model. (e)-(h) GraphSAGE model.

baseline model. In the case of the GCN network, SN exhibits slightly faster training times than our method, and takes, on average, $1.12 \times$ the time needed to train the baseline model per epoch.

Table 9 records the average total time for training each network. As AT is being performed starting from a pre-trained network, the total training time is represented by the time needed for the conventional training (Baseline GNN), the time required to generate the adversarial samples (AT attack), and the actual training time with the adversarial input (AT). Because training the network subjected to a Lipschitz constraint does

not require starting from a pre-trained network, our method takes less time to train a model, on average, than AT. As expected, RS is comparable to the baseline GNN in terms of training time. However, we need to take into account its major drawbacks. To achieve decent robustness, RS requires performing inference multiple times, which results in extensive time consumption, particularly for larger graphs (as shown in Table 10). In our experiments, RS took 150 to 400 times longer for inference compared to the other methods. This might render RS impractical for real-life applications, as it takes several seconds for a single inference. In contrast, our method requires minimal time for inference, making it a more efficient choice.

Table 8: Average training time ± standard deviation per epoch (GPU) in milliseconds.

| | Dataset | Baseline GNN | Constrained GNN | RS | AT | SN |
|---|---|---|---|---|---|---|
| GCN | Facebook | 17.7±0.2 | 24.2±0.2 | 19.2±0.7 | 18.4±0.5 | 19.7±0.4 |
| | GitHub | 17.6±0.2 | 23.3±0.4 | 19.9±2.0 | 18.3±0.4 | 19.4±0.3 |
| | LastFM Asia | 17.5±0.4 | 27.7±0.2 | 17.4±0.3 | 17.6±0.2 | 20.0±0.5 |
| | Deezer Europe | 17.4±0.3 | 23.2±0.4 | 17.5±0.5 | 17.9±0.4 | 19.6±0.2 |
| Graph SAGE | Facebook | 19.5±0.2 | 26.3±0.3 | 18.3±0.9 | 20.0±0.5 | - |
| | GitHub | 21.4±0.3 | 27.0±0.2 | 19.8±2.1 | 21.9±0.3 | - |
| | LastFM Asia | 18.3±0.3 | 28.8±0.3 | 18.3±0.3 | 18.5±0.2 | - |
| | Deezer Europe | 18.7±0.2 | 24.6±0.3 | 18.9±0.3 | 19.4±0.4 | - |

Table 9: Average total training time ± standard deviation in seconds (GPU). *AT attack* represents the time to perform APGD attack with DL loss; *AT total* represents the total time for performing adversarial training.

| | Dataset | Baseline GNN | Constrained GNN | RS | AT | AT attack | AT total | SN |
|---|---|---|---|---|---|---|---|---|
| GCN | Facebook | 12.0±0.8 | 22.0±8.2 | 11.9±1.0 | 9.5±0.9 | 1.2±0.1 | 22.7±1.2 | 15.6±0.9 |
| | GitHub | 18.7±2.2 | 15.5±2.0 | 14.8±5.0 | 13.9±4.0 | 1.3±0.0 | 33.9±4.6 | 14.2±2.0 |
| | LastFM Asia | 21.1±4.0 | 45.4±9.4 | 21.7±8.6 | 9.0±1.7 | 1.0±0.0 | 31.1±4.4 | 28.2±1.7 |
| | Deezer Europe | 12.9±3.1 | 15.7±2.3 | 10.4±1.8 | 9.3±0.7 | 1.1±0.0 | 23.2±3.2 | 13.9±0.9 |
| Graph SAGE | Facebook | 12.4±0.7 | 20.0±9.4 | 16.0±4.0 | 11.9±1.0 | 1.9±0.1 | 26.2±1.2 | - |
| | GitHub | 11.5±1.9 | 18.7±1.7 | 17.8±8.5 | 10.0±0.4 | 2.6±0.0 | 24.2±2.0 | - |
| | LastFM Asia | 12.5±1.8 | 44.9±7.5 | 10.4±0.7 | 8.8±0.2 | 1.2±0.0 | 22.5±1.8 | - |
| | Deezer Europe | 9.4±0.3 | 17.4±5.1 | 9.3±0.5 | 9.0±0.4 | 1.5±0.0 | 19.9±0.5 | - |

Table 10: Average inference time ± standard deviation in milliseconds for different models (GPU).

| | Dataset | Baseline GNN | Constrained GNN | RS | AT | SN |
|---|---|---|---|---|---|---|
| GCN | Facebook | 14.8±2.3 | 13.7±3.9 | 4275.9±242.2 | 14.8±1.2 | 14.6±2.9 |
| | GitHub | 16.1±4.2 | 16.3±4.4 | 6268.2±228.6 | 16.3±4.2 | 16.1±4.1 |
| | Deezer Europe | 14.4±0.3 | 15.0±0.8 | 4848.8±247.6 | 14.4±1.2 | 14.8±0.6 |
| | LastFM Asia | 12.3±4.1 | 10.7±3.3 | 2536.0±277.7 | 10.3±0.3 | 11.0±3.7 |
| Graph SAGE | Facebook | 16.0±1.3 | 17.9±3.4 | 5099.6±230.7 | 15.9±1.9 | - |
| | GitHub | 19.0±1.2 | 18.6±3.9 | 7795.1±247.1 | 18.6±2.9 | - |
| | Deezer Europe | 16.1±0.5 | 14.7±0.4 | 5376.9±218.6 | 14.6±4.2 | - |
| | LastFM Asia | 12.5±0.4 | 11.7±3.0 | 2614.6±175.3 | 10.5±0.7 | - |

## D.5 Impact of the number of layers on robustness

We consider the two particular GNN formulations, symmetrical GraphSAGE and GCN, for which we train networks of varying depths $m \in \{2, 4, 6, 7\}$. The hidden dimension of each layer is $N_i = 16$ for $i \in \{1, \ldots, m\}$.

We observe that the deeper the network, the less robust it is to adversarial perturbations. The example of the Facebook dataset is shown in Figure 7. As the number of layers increases, the initial Lipschitz constant of the network is higher, and thus, it is harder to impose a tight Lipschitz bound. However, we can still find a Lipschitz bound that gives a good robustness accuracy trade-off as the number of layers increases.

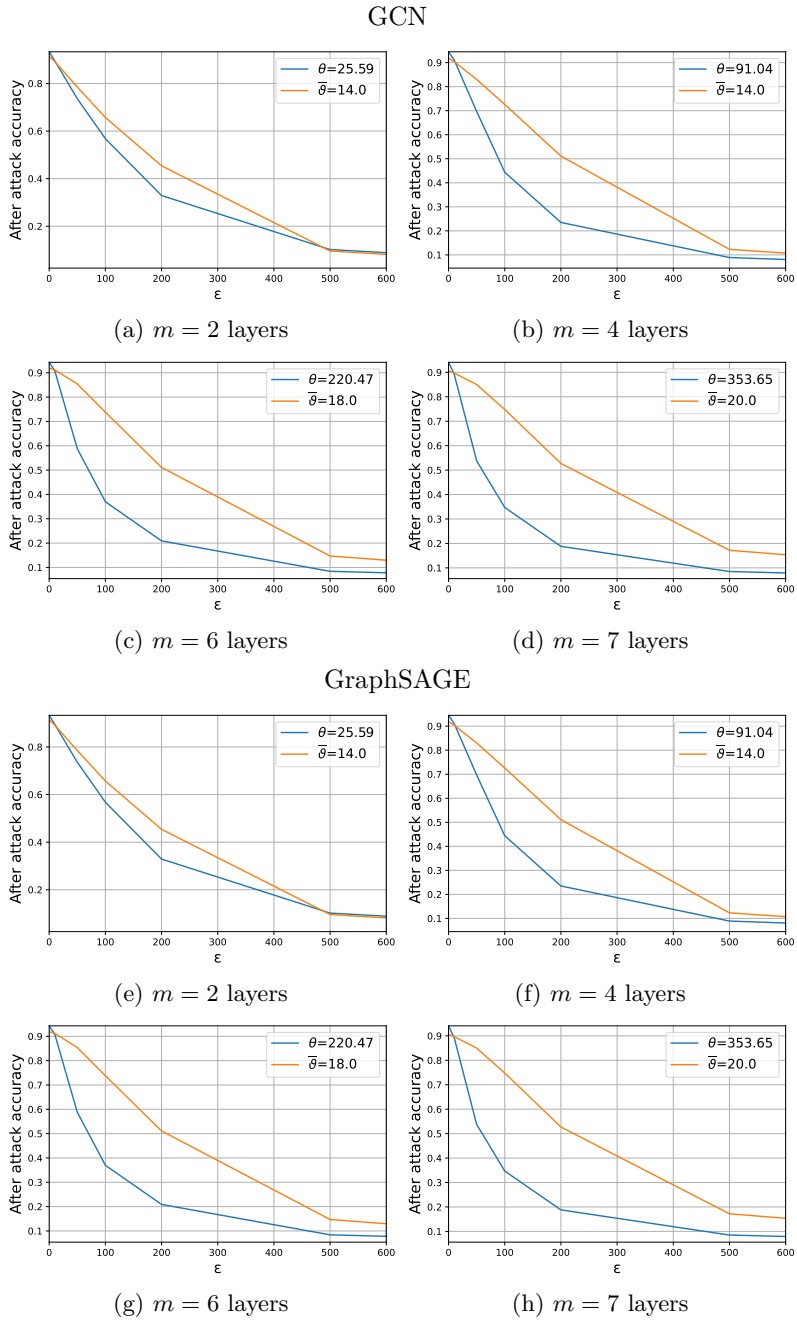

Figure 7: Impact of the number of layers on robustness for the Facebook dataset. Accuracy vs. $\ell_2$ norm of the test set perturbation ($\epsilon$). APGD attack with DL loss. (a)-(d) GCN model. (e)-(h) GraphSAGE model

