# OpenReview forum: "Training Graph Neural Networks Subject to a Tight Lipschitz Constraint"
_TMLR — Accepted by TMLR_

### Review · Reviewer_EkfX · 2024-02-28

**Summary Of Contributions:**

This paper proposes a new method of estimating the Lipschitz constant for MPNN-type GNNs whose activation functions are Lipschitz continuous and whose weights are non-negative. This estimation is tight when the activation function is ReLU. The proposed method includes GraphSAGE and GCN as special cases and specific estimates of the Lipschitz constant are derived for those cases.
Based on the proposed estimation, this paper also proposes an algorithm for robust training in which the Lipschitz constant of the model is constrained to a given value.
Numerical experiments apply white box global adversarial attacks that modify node features (PGD and APGD) to GNNs trained by the proposed method. The accuracy and robustness are evaluated using multiple datasets and compared with multiple baselines, including robust training methods (adversarial training, Randomized Smoothing), regularization methods (spectral normalization), and robust training for adversarial attacks that change the topology of the input graph (GCN-Jaccard, SVD-GCN, RGCN).

**Audience:**

Yes

**Broader Impact Concerns:**

No broader impact concerns.

**Claims And Evidence:**

Yes

**Requested Changes:**

Q.1 This paper claims that the first of the robust training against GNNs is to address the case of networks with continuous node features and fixed topology. However, robust training methods for usual FNNs are defense methods against adversarial attacks that change node features. I want to clarify whether we can use them as baseline methods.


Q.2 In Algorithm 1, does DFB stand for dual forward-backward? If so, write it explicitly.


Q.3 Section 4.1 writes:

> However, in our experiments, we consider the same bias vector, which is the standard practice.

Does this mean numerical experiments use the same learnable bias vector for all layers? I want to clarify the supporting evidence because I do not know such a practice if my understanding is correct.


Q.4 Section 4.1 writes:

> As we tighten the constant, the drop in accuracy becomes more significant, but the model's ability to withstand adversarial pertubations improves compared to the conventionally trained model

However, I could not find the rationale for this. Table 2 shows that the accuracy is increased. However, it was not clear whether lowering θ improves robustness. I want to clarify this point.


Q.5 In Section 4.2, this sentence was hard to parse:

> Comparing the performance of the PGD and APGD attacks with Cross Entropy [...] and Difference of Logitis [...] losses, and Differences of Logits [...] kisses

Does this mean that six different attack methods were compared by combining two attack methods (PGD and APGD) and three losses (CE, DL, DLR)?

**Strengths And Weaknesses:**

Strength
- The writing is clear. I can understand the main points of the paper without difficulties.
- As far as I have confirmed, the theoretical analysis, specifically the proof of Theorem 3.1, is correct.
- It is numerically verified that the proposed method is robust against adversarial attacks (Figure 2).
- The proposed estimation method has the limitation that it can only be applied when the weights are positive. This paper properly verifies that this limitation does not significantly impact performance (Table 2).
- The limitation is discussed correctly (Section 6).
- Audience is appropriate. In the research of social networks, it is important to study protection against adversarial attacks, such as preventing the propagation of fake news and misinformation. Therefore, robust training of GNNs is an important research topic that interests the TMLR audience.

Weakness
- This paper claims that the lower the Lipschitz constant is, the better the robustness. However, if I get all the information, no such experimental results are shown (ref: Q.4).

---

> ### Author Response · Authors · 2024-03-14
> **Acknowledgement of Reviewer Feedback and Revisions Made**
>
> We thank the reviewer for the time and effort invested in reviewing our manuscript. We appreciate the positive feedback and wish to address the questions and weaknesses pointed out by the reviewer.
>
> **Requested Changes Q1:** Robust training methods for usual FNNs can theoretically be used against attacks that change node features. As shown in subsection *4.3 Memory and computational efficiency*, standard methods for controlling the Lipschitz constant of FNNs are inefficient when applied to GNNs. Other methods that we use in our comparisons like adversarial training and randomized smoothing are also adaptations of robust training techniques for FNNs.
>
> **Requested Changes Q2:** Indeed, DFB stands for dual forward-backward. Thank you for this observation, we have made the abbreviation explicit in the revised version of the manuscript.
>
> **Requested Changes Q3:** We use a different learnable bias vector at each layer. What we meant by this observation is that our formulation in Equation (1) allows the bias vector to be different at each node of the graph ($b_v^{(i)}$). This would involve having a distinctive bias parameter for each node and incorporating it into the computation for those specific node features. This approach could potentially increase the expressiveness of the model by allowing node-specific biases, but it could lead to overfitting.
>
> In both the standard formulations of the graph convolutional operator (Kipf \& Welling, 2016) and the GraphSAGE model (Hamilton et al., 2017a), a single bias vector is used for the entire layer and applied element-wise to the features of all nodes. The bias term is not node-specific in these formulations. This is the way we define and apply the bias vector in our experiments.
>
> We have reformulated this remark in the revised version of the paper: "The general formulation in Equation (1) allows for a distinctive bias parameter for each node. However, in our experiments, a single bias vector is used for the entire layer and applied element-wise to the features of all nodes, which is the standard practice.".
>
> **Requested Changes Q4:** In Table 2, we observe that decreasing the imposed Lipschitz constant on the model results in a reduction in accuracy. For instance, the accuracy of the model when imposing a Lipschitz constant of $\overline{\vartheta}=30$ is 93.1\%, whereas enforcing a constraint of $\overline{\vartheta}=15$ leads to a decrease in the initial accuracy to 91.7\%.
>
> To provide a clearer illustration of this trade-off, we have introduced *Figure 1: Accuracy - robustness tradeoff* in *Section 4.2* of the revised manuscript.  This figure demonstrates the relationship between accuracy and robustness on different datasets, showcasing the accuracy of both baseline and constrained models under various $\ell_2$ norm of the test set perturbation levels. As we lower the value of $\overline{\vartheta}$, the robustness of the model improves, but this enhancement comes at the cost of a greater initial drop in accuracy compared to the baseline model. The green plot in the figure represents a potential trade-off between these two factors. We trust that this visualization provides a more explicit depiction of the accuracy-robustness relationship.
>
> **Requested Changes Q5:** Indeed, your understanding is correct. We compared six different attack methods formed by combining two attack techniques (PGD and APGD) with three distinct loss functions (Cross-Entropy (CE), Difference of Logits (DL), and Difference of Logits Ratio (DLR)).
>
> To enhance clarity, we have revised the sentence in *Section 4.2* of the paper: "In evaluating a total of six attack methods, formed by combining two attack techniques (PGD and APGD) with three distinct loss functions (Cross-Entropy (CE), Difference of Logits (DL), and Difference of Logits Ratio (DLR)  (Carlini \& Wagner, 2017; Croce \& Hein, 2020)), we consistently found that the APGD attack outperformed PGD irrespective of the chosen loss function. Furthermore, regardless of the attack method, the DL loss demonstrated the strongest adversarial impact. Specifically, the APGD attack coupled with the DL loss emerged as the most powerful among the six attacks, across all datasets.".
> We hope this revision provides a clearer description of the evaluation setup.
>
>
> We hope that we have addressed all the concerns raised by the reviewer. We appreciate the constructive suggestions, and we believe that the changes we have implemented have enhanced the quality and clarity of our manuscript.

---

> > ### Comment · Reviewer_EkfX · 2024-03-20
> > **Response**
> >
> > I thank the authors for answering my questions. All of my questions are now clear.
> >
> > - Q.1 OK
> > - Q.2 OK
> > - Q.3 OK. I agree with the authors that using the same bias vector for all nodes is a standard practice.
> > - Q.4 OK. I understand that Figure 1 justifies the claim.
> > - Q.5 OK.

---

### Review · Reviewer_HsXc · 2024-03-01

**Summary Of Contributions:**

In this paper, the author analyze the Lipschitz constant of Graph Neural Networks with nonnegative weights (the proof is based on an adaptation of a particular case of a paper by Combette and Pesquet.), and develop an optimization method to train these GNNs while enforcing constraints on this Lipschitz constant (based on projected block gradient descent). Motivated by robustness to adversarial attacks on the node features, they perform experiments to showcase the performance and stability of their method compared to baselines, focusing on the popular GCN and GraphSage architecture.

**Audience:**

Yes

**Claims And Evidence:**

Yes

**Requested Changes:**

See above. A thorough discussion, and at the very least more experiments, around the nonnegativity constraints, which seems very constraining.

**Strengths And Weaknesses:**

Strengths:
- the paper is generally well-written and easy to follow
- the topic is interesting, the proposed solution is clever and seems to perform well on the selected datasets
- the paper and appendices are quite complete

Weaknesses:
- the major weakness in my opinion is the nonnegativity constraint on the weights. As pointed out by the authors, this does not seem to affect *the selected datasets*, but it is very probable that it is a **very** strong constraint otherwise. In particular, when using ReLU activations as done in the experiments, after the first layer the outputs are themselves nonnegative, so the subsequent ReLU will never be activated, and the GNN is essentially linear!! (in which case the Lipschitz constant is indeed trivially tight). Linear GNNs might still perform okay (see the SGC model), but the contribution is substantially different in this case. At the very least, other activation functions should be tried (sigmoid, tanh, or even non-conventional ones that output negative values for positive inputs, to emphasize non-trivial interactions with nonnegative weights).
- again regarding nonnegativity, the paper by Combette and Pesquet is quite extensive, and nonnegative NNs are only a small particular case, could you see other results from this paper be adapted to GNNs? If not, why?
- a discussion on the proof of the theorem would be appreciable, in particular, how different is it from the original proof? Is this "just" adapted to the particular case of GNNs (which is absolutely fine), or are there intrinsic difficulties due to GNNs?

Minor remark:
- the presentation of the GNNs is quite verbose, with many notations and matrices for simple models. This might be improved, leaving more space for discussions

---

> ### Author Response · Authors · 2024-03-14
> **Acknowledgement of Reviewer Feedback and Revisions Made**
>
> We would like to express our appreciation to the reviewer for his/her time and effort in evaluating our manuscript and for the constructive feedback. We subsequently address the weaknesses highlighted by the reviewer.
>
> **Weakness 1:** We emphasize that, while the weights within the network are nonnegative, the inclusion of signed biases introduces arbitrary signs in the outputs after each layer. Consequently, this allows for the activation function (even ReLU) to be triggered. We have explicitly addressed this in the revised version of the manuscript.
>
> Moreover, following the reviewer’s suggestion, we explored four other alternative activation functions, namely sigmoid, hyperbolic tangent (tanh), leaky ReLU, and sigmoid linear unit (SiLU). Our findings consistently demonstrated that our models outperform the baseline, showcasing robustness against adversarial attacks. These additional results have been included in *Appendix D.3* of the revised paper.
>
> **Weakness 2:** Regarding the nonnegativity aspect, we acknowledge the extensive work by Combettes and Pesquet (2020). While it is conceivable that some results from their paper could be adapted to GNNs, particularly in exploring alternative norms beyond the Euclidean one, it must be emphasized that such adaptations are not straightforward and may not be easily demonstrable. We have highlighted this potential perspective in the conclusion of the revised manuscript. Concerning NNs with weights of arbitrary signs, the work by Combettes and Pesquet leads to bounds which are tight, but costly in terms of computations.
>
> **Weakness 3:** Our proof for the theorem differs significantly from the one presented in Combettes and Pesquet (2020). We do not aim to re-prove any results from their article; rather, we leverage their findings to derive new results specifically tailored for GNNs.
> The main challenge lies in expressing the conditions outlined in Proposition 5.10 of this paper in a way that makes them easily applicable to GNNs from a computational viewpoint. This challenge is addressed in our work, with a focus on achieving substantial reductions in memory requirements, as detailed in *Section 4.3. Memory and Computational Efficiency*.
> Our proofs heavily rely on graph spectral theory, a point that is explicitly demonstrated in *Appendix A* of our manuscript. We believe that our approach enhances the computational feasibility of applying the theoretical conditions to GNNs.
>
> **Minor remark:** We have condensed the introduction to GNN models by simplifying explanations and notation. These adjustments have been incorporated into the revised version of the manuscript, which is now available for a second review.
>
> **Requested Changes:** In response to the reviewer's request for changes, we hope that the pointed-out weaknesses have been addressed properly. We have described the nonnegativity constraints more comprehensively and conducted additional experiments on alternative activation functions. We appreciate the feedback provided by the reviewer, and we believe these changes contributed to the improvement of our manuscript.

---

### Review · Reviewer_sCZQ · 2024-03-07

**Summary Of Contributions:**

The paper describes a tighter, non-separable bound for the Lipschitz constant of a graph neural network, and then proposes a projected subgradient approach for constrained optimization of such graph neural networks. Experiments that evaluate the efficacy of such an approach are provided. The authors motivated the approach from a security and robustness perspective.

**Audience:**

Yes

**Broader Impact Concerns:**

No ethical concerns.

**Claims And Evidence:**

Yes

**Requested Changes:**

1. Lipschitz constants of a function/class of function is not unique. A 1-Lipschitz function is also 2-Lipschitz. What the authors show in theorem 3.1 is a upper bound on Lipschitz constants. As such, words like "the Lipschitz constant" (which appears frequently, including in the abstract) is confusing. This wording confusion is furthered by the author's tendency to mix "the Lipschitz constant" with "a Lipschitz constant" together, for example in page 5 of section 3.2. If the authors are referring to the smallest lipschitz constant possible for a given function, I suggest that they can use "the optimal Lipschitz constant" in the text, like in theorem 3.1, to avoid confusion.

**Strengths And Weaknesses:**

Strengths:

1. Clarity: the article is written clearly, in an accessible and easy to read manner.
2. Relevance: the authors motivated their work well, and the topic is of interest to at least some TMLR readers.
3. Correctness: the theoretical claims are, to the best of my knowledge (I did not check the supplementary materials line by line), correct and reasonable.
4. Promising experimental results.

Weaknesses:

1. Wording: Certain wording of the phrases/text makes the material appear unclear. I outline such requested changes in the section below.

---

> ### Author Response · Authors · 2024-03-14
> **Refinement of Wording in Response to Reviewer Feedback**
>
> We would like to thank the reviewer for his/her time and effort in evaluating our manuscript, and for the constructive remark. Following the reviewer’s feedback, we have revised the wording throughout the paper. We now use the term ”a” Lipschitz constant” when referring to a bound on ”the” (optimal) Lipschitz constant corresponding to the smallest possible Lipschitz constant for a model. However, in some sentences, we found it necessary to use ”the Lipschitz constant” in a general meaning to make them grammatically correct. These adjustments have been incorporated into the newly uploaded revised version of the paper.

---

### Decision · Action_Editor_1sCu · 2024-04-17

**Recommendation:** Accept as is

**Comment:**

The reviewers found the paper well-written, interesting, and correct. They mostly had only minor concerns about it, many of which have been addressed in the revised version of the paper. It is not clear how limiting the constraint for the weights to be non-negative is in general, but the authors are upfront about this and the results in the paper show that it does not degrade model performance substantially on a range of datasets.

**Audience:**

Estimating Lipschitz constants of GNNs and defending GNNs against adversarial attacks are topics of considerable interest to the TMLR audience.

**Claims And Evidence:**

The authors propose a new approach of estimating Lipschitz constants for Graph Neural Networks with non-negative weights and Lipschitz-continuous activation functions, with the estimates being tight for networks with ReLU activations. They also introduce an algorithm for training GNNs while constraining their Lipschitz constant to a pre-specified value, and demonstrate its effectiveness at defending GNNs from adversarial attacks targeting node features. The derivations appear to be correct and the empirical evaluation is convincing.